# Antagonism between *germ cell-less* and Torso receptor regulates transcriptional quiescence underlying germline/soma distinction

Megan M Colonnetta[1], Lauren R Lym[2], Lillian Wilkins[1], Gretchen Kappes[1], Elias A Castro[2], Pearl V Ryder[2†], Paul Schedl[1], Dorothy A Lerit[2]*, Girish Deshpande[1]*

[1]Department of Molecular Biology, Princeton University, Princeton, United States; [2]Department of Cell Biology, Emory University School of Medicine, Atlanta, United States

**\*For correspondence:**
dlerit@emory.edu (DAL);
gdeshpan@princeton.edu (GD)

†ORCID: 0000-0003-3699-3633

**Competing interests:** The authors declare that no competing interests exist.

**Abstract** Transcriptional quiescence, an evolutionarily conserved trait, distinguishes the embryonic primordial germ cells (PGCs) from their somatic neighbors. In *Drosophila melanogaster*, PGCs from embryos maternally compromised for *germ cell-less* (*gcl*) misexpress somatic genes, possibly resulting in PGC loss. Recent studies documented a requirement for Gcl during proteolytic degradation of the terminal patterning determinant, Torso receptor. Here we demonstrate that the somatic determinant of female fate, *Sex-lethal* (*Sxl*), is a biologically relevant transcriptional target of Gcl. Underscoring the significance of transcriptional silencing mediated by Gcl, ectopic expression of a degradation-resistant form of Torso (*torso$^{Deg}$*) can activate *Sxl* transcription in PGCs, whereas simultaneous loss of *torso-like* (*tsl*) reinstates the quiescent status of *gcl* PGCs. Intriguingly, like *gcl* mutants, embryos derived from mothers expressing *torso$^{Deg}$* in the germline display aberrant spreading of pole plasm RNAs, suggesting that mutual antagonism between Gcl and Torso ensures the controlled release of germ-plasm underlying the germline/soma distinction.

## Introduction

Following fertilization, a Drosophila embryo undergoes 14 consecutive nuclear divisions to give rise to the cellular blastoderm. While the initial nuclear divisions take place in the center of the embryo, the nuclei begin to migrate toward the periphery around nuclear cycle (NC) 4–6 and reach the cortex at NC9/10 (*Farrell and O'Farrell, 2014*). Even before bulk nuclear migration commences, a few nuclei move toward the posterior of the embryo, enter a specialized, maternally derived cytoplasm known as the pole plasm, and induce the formation of pole buds (PBs) (*Williamson and Lehmann, 1996*; *Wilson and Macdonald, 1993*; *Wylie, 1999*). The centrosomes associated with these nuclei trigger the release of pole plasm constituents from the posterior cortex and orchestrate precocious cellularization to form the primordial germ cells (PGCs), the progenitors of the germline stem cells in adult gonads (*Lerit and Gavis, 2011*; *Raff and Glover, 1989*). Unlike pole cell nuclei, somatic nuclei continue synchronous divisions after they reach the surface of the embryo until NC 14 when they cellularize (*Blythe and Wieschaus, 2015*).

The timing of cellularization is not the only difference between the soma and PGCs. Although newly formed PGCs divide after they are formed, they undergo only one or two asynchronous divisions before exiting the cell cycle. Another key difference is in transcriptional activity (*Nakamura and Seydoux, 2008*). Transcription commences in the embryo during NC 6–7 when a select number of genes are active (*Ali-Murthy et al., 2013*). Transcription is more globally

upregulated when the nuclei reach the surface, and by the end of NC 14, zygotic genome activation (ZGA) is complete (*Ali-Murthy et al., 2013*; *Harrison and Eisen, 2015*). This transition is marked by high levels of phosphorylation of residues Serine 5 (Ser5) and Serine 2 (Ser2) in the C-terminal domain (CTD) of RNA polymerase II (*Schaner et al., 2003*; *Seydoux and Dunn, 1997*). By contrast, in newly formed PGCs, transcription is switched off, and PGC nuclei have only residual amounts of Ser5 and Ser2 CTD phosphorylation (*Deshpande, 2004*; *Martinho et al., 2004*; *Seydoux and Dunn, 1997*). Moreover, and consistent with their transcriptionally quiescent status, other changes in chromatin architecture that accompany ZGA are also blocked in PGCs (*Schaner et al., 2003*).

Three different genes, *nanos* (*nos*), *polar granule component* (*pgc*), and *germ cell-less* (*gcl*), are known to be required for establishing transcriptional quiescence in newly formed PGCs (*Deshpande et al., 2005*; *Deshpande, 2004*; *Deshpande et al., 1999*; *Hanyu-Nakamura et al., 2008*; *Kobayashi et al., 1996*; *Leatherman et al., 2002*; *Martinho et al., 2004*). The PGCs in embryos derived from mothers carrying mutations in these genes fail to inhibit transcription, and this compromises germ cell specification and disrupts germ cell migration. (As these are maternal effect genes, embryos derived from *nos/pgc/gcl* mothers display the resulting mutant phenotypes and will be referred to as *nos/pgc/gcl* here onwards.) Interestingly, these three genes share only a few targets, suggesting overlapping yet distinct mechanisms of action. Nos is a translation factor and thus must block transcription indirectly. Together with the RNA-binding protein Pumilio (Pum), Nos interacts with recognition sequences in the 3'-untranslated regions (3'UTRs) of mRNAs and inhibits their translation (*Asaoka et al., 2019*; *Sonoda and Wharton, 1999*; *Wharton and Struhl, 1991*). Currently, the key mRNA target(s) that Nos-Pum repress to block transcription is unknown; however, in *nos* and *pum* mutants, PGC nuclei display high levels of Ser5 and Ser2 CTD phosphorylation and activate transcription of gap and pair-rule patterning genes and the sex determination gene *Sex-lethal* (*Sxl*) (*Deshpande et al., 2005*; *Deshpande et al., 1999*). *pgc* encodes a nuclear protein that binds to the transcriptional elongation kinase p-TEFb, blocking Ser5 CTD phosphorylation (*Hanyu-Nakamura et al., 2008*). In *pgc* mutant pole cells, Ser5 phosphorylation is enhanced, as is transcription of several somatic genes, including genes involved in terminal patterning (*Deshpande, 2004*; *Martinho et al., 2004*).

While the primary function of *nos* and *pgc* appears to be blocking ZGA in PGCs, *gcl* has an earlier function, which is to turn off transcription of genes activated in somatic nuclei prior to nuclear migration (*Leatherman et al., 2002*). Targets of *gcl* include two X-chromosome counting elements (XCEs), *scute* (*sc/sis-b*) and *sisterless-a* (*sis-a*), that function to turn on the sex determination gene, *Sxl*, in female soma (*Cline and Meyer, 1996*; *Salz and Erickson, 2010*). *gcl* embryos not only fail to shut off *sis-a* and *sis-b* transcription in PBs, but also show disrupted PGC formation. In some *gcl* embryos, PGC formation fails completely, while in other embryos only a few PGCs are formed (*Cinalli and Lehmann, 2013*; *Jongens et al., 1992*; *Lerit et al., 2017*; *Robertson et al., 1999*). In this respect, *gcl* differs from *nos* and *pgc*, which have no effect on the process of PGC formation, but instead interfere with the specification of PGC identity.

Studies by *Leatherman et al., 2002* suggested that the defects in PGC formation in *gcl* mutant embryos are linked to failing to inhibit somatic transcription. They found that when PBs first form during NC 9 in wild-type (WT) embryos, levels of CTD phosphorylation PB are only marginally less than in nuclei elsewhere in the embryo. However, by NC 10, there was a dramatic reduction in CTD phosphorylation even before PBs cellularize. By contrast, in *gcl* mutant embryos, about 90% of the NC 10 PB nuclei had CTD phosphorylation levels approaching that of somatic nuclei. Moreover, this number showed an inverse correlation with the number of PGCs in blastoderm stage *gcl* embryos. Whereas WT blastoderm embryos have >20 PGCs per embryo, *gcl* embryos had on average just three PGCs under their culturing conditions. Interestingly, expression of the mouse homologue of Gcl protein, mGcl-1, can rescue the *gcl* phenotype in *Drosophila* (*Leatherman et al., 2000*). Supporting the conserved nature of the involvement of Gcl during transcriptional suppression, a protein complex between mGcl-1 and the inner nuclear membrane protein LAP2β is thought to sequester E2F:D1 to reduce transcriptional activity of E2F:D1 (*Nili et al., 2001*).

The connection Leatherman et al. postulated between failing to turn off ongoing transcription and defects in PGC formation in *gcl* mutants is controversial and unresolved. This model predicts that a non-specific inhibition of polymerase II should be sufficient to rescue PGC formation in *gcl* embryos. However, *Cinalli and Lehmann, 2013* found that the PGC formation defects seen in *gcl* embryos were not rescued after injection of the RNA polymerase inhibitor, α-amanitin. Since α-

amanitin treatment disrupted somatic cellularization without impacting PGC formation in WT embryos, they concluded that it effectively blocked polymerase transcription. On the other hand, subsequent experiments by *Pae et al., 2017* raised the possibility that inhibiting transcription in pole cell nuclei is a critical step in PGC formation. These authors showed that Gcl is a substrate-specific adaptor for a Cullin3-RING ubiquitin ligase that targets the terminal pathway receptor tyrosine kinase, Torso, for degradation. The degradation of Torso would be expected to prevent activation of the terminal signaling cascade in PGCs. In the soma, Torso-dependent signaling activates the transcription of several patterning genes, including *tailless,* that are important for forming terminal structures at the anterior and posterior of the embryo (*Casanova and Struhl, 1989*; *Klingler et al., 1988*; *Martinho et al., 2004*; *Pignoni et al., 1992*; *Strecker et al., 1989*). Thus, by targeting Torso for degradation, Gcl would prevent the transcriptional activation of terminal pathway genes by the MAPK/ERK kinase cascade in PGCs. Consistent with this possibility, simultaneous removal of *gcl* and either the Torso ligand modifier, *torso-like* (*tsl*) or *torso* resulted in rescue of germ cell loss induced by *gcl*. Surprisingly, however, *Pae et al., 2017* were unable to observe a similar rescue of *gcl* phenotype when they used RNAi knockdown to compromise components of the MAP kinase cascade known to act downstream of the Torso receptor (*Ambrosio et al., 1989*; *Duffy and Perrimon, 1994*; *Furriols and Casanova, 2003*). Based on these findings, they proposed that activated Torso must inhibit PGC formation via a distinct non-canonical mechanism that is both independent of the standard signal transduction pathway and does not involve transcriptional activation.

In the studies reported here, we have revisited these conflicting claims by examining the role of Gcl in establishment/maintenance of transcriptional quiescence. The studies of *Leatherman et al., 2002* indicated that two of the key X chromosomal counting elements, *sis-a* and *sis-b*, were inappropriately expressed in *gcl* PBs and PGCs. Since transcription factors encoded by these two genes function to activate the *Sxl* establishment promoter, *Sxl-Pe*, in somatic nuclei of female embryos, their findings raised the possibility that *Sxl* might be ectopically expressed in PBs/PGCs of *gcl* embryos. Here we show that in *gcl* embryos, *Sxl* transcription is indeed inappropriately activated in PBs and newly formed PGCs. Moreover, ectopic expression of *Sxl* in early embryos disrupts PGC formation similar to *gcl*. Supporting the conclusion that *Sxl* is a biologically relevant transcriptional target of Gcl, PGC formation defects in *gcl* embryos can be suppressed either by knocking down *Sxl* expression using RNAi or by loss-of-function mutations. As reported by *Pae et al., 2017*, we found that loss of *torso-like* (*tsl*) in *gcl* embryos suppresses PGC formation defects. However, consistent with a mechanism that is tied to transcriptional misregulation, rescue is accompanied by the reestablishment of transcriptional silencing in *gcl* PGCs. Lending further credence to the idea that transcription misregulation plays an important role in disrupting PGC development in *gcl* embryos, we found that expression of a mutant form of Torso that is resistant to Gcl-dependent degradation (hereafter referred to as *torso*^Deg^ *Pae et al., 2017*) ectopically activates transcription of two Gcl targets, *sis-b* and *Sxl*, in PB and PGC nuclei. In addition, stabilization of Torso in early PGCs also mimics another *gcl* phenotype, the failure to properly sequester key PGC determinants in PBs and newly formed PGCs.

## Results

### Gcl represses the expression of XCEs in nascent PGCs

To reexamine the role of *gcl* in transcriptional quiescence reported by *Leatherman et al., 2002*, we first used single molecule fluorescent in situ hybridization (smFISH) to assess whether *sis-b* is properly turned off in *gcl* mutants. As shown in *Figure 1a*, nuclear *sis-b* transcripts are not detected in WT PBs or PGCs (n = 16 embryos). In contrast, in 67% of *gcl* embryos, we observed *sis-b* transcripts in PB and PGC nuclei (*Figure 1b*, n = 21 embryos, p=2.1e-05). *sis-b* transcripts are present most frequently in *gcl* PBs; however, we can also detect transcripts after PGC cellularization. Leatherman et al. reported that a second XCE, *sis-a*, is not properly turned off in *gcl* PBs and PGCs. To determine if *gcl* is required to repress other XCEs besides *sis-a* and *sis-b*, we probed for *runt* expression in *gcl* mutants. We found that like *sis-b*, *runt* is also expressed in a subset of *gcl* PB nuclei and PGCs (29% of *gcl* embryos, n = 11, p=0.009653), while it is never observed in WT PBs or PGCs. Curiously, in experiments where we examined *sis-b* and *runt* transcription simultaneously, we observed some PB/PGC nuclei that expressed both XCEs, and some that only expressed one or the other. In this

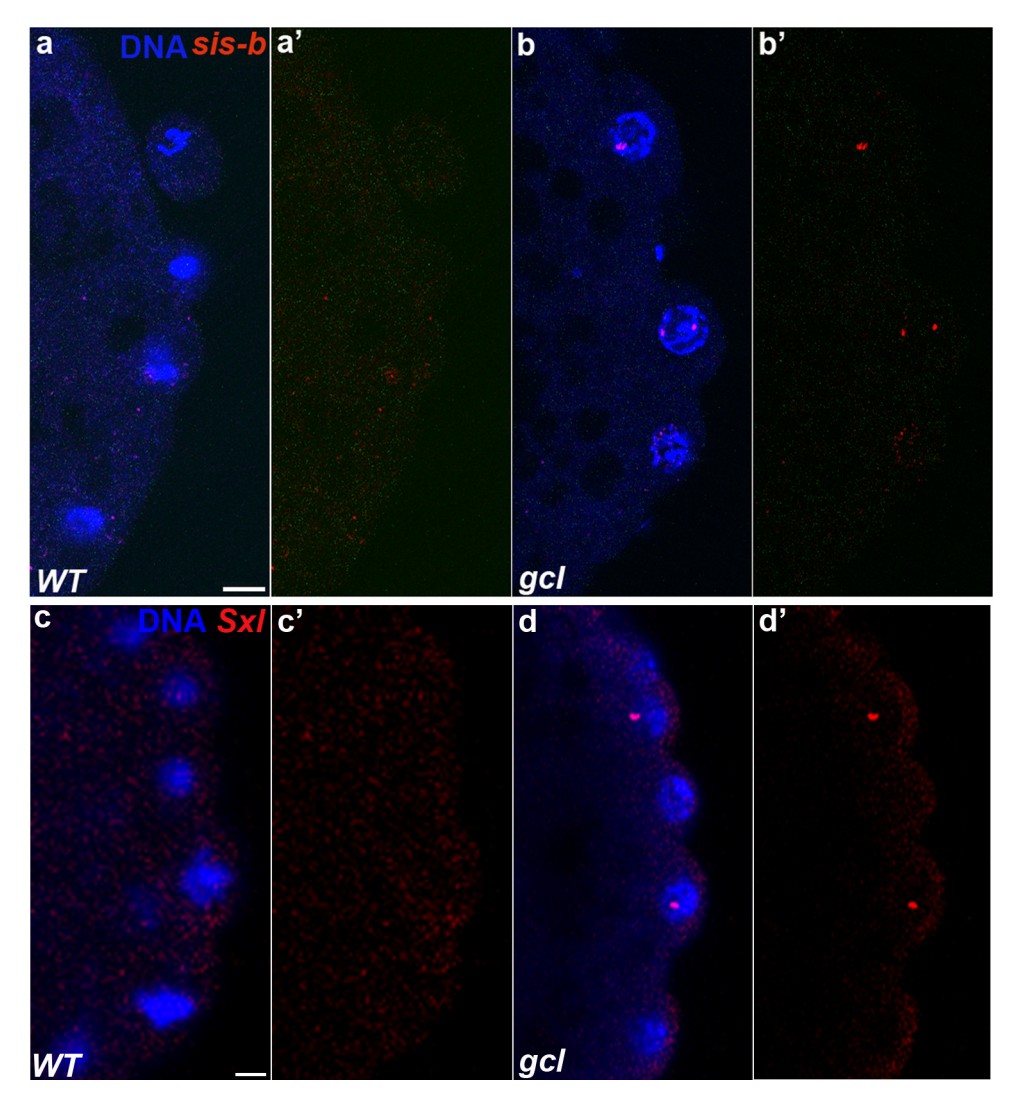

**Figure 1.** *sis-b* and *Sxl* are transcribed in *gcl* PBs and PGCs. smFISH was performed using probes specific for *sis-b* or *Sxl* on 0–3 hr old embryos to assess the status of transcription in *gcl* PBs. Wild-type (WT) embryos of similar age were used as control. Posterior poles of representative pre-syncytial blastoderm embryos are shown with *sis-b* (**a/b**) or *Sxl* (**c/d**) RNA visualized in red and Hoescht DNA dye in blue. While 0% of control embryos display *sis-b* (**a/a'**, n = 16) or *Sxl* (**c/c'**, n = 18) transcription in PBs, transcription of both *sis-b* (**b/b'**) and *Sxl* (**d/d'**) is detected in *gcl* mutant PBs. We observed *sis-b* transcription in 67% (n = 21, p=2.10e-05) and *Sxl* transcription in 42% (n = 31, p=0.001593) of *gcl* embryos. Scale bar represents 10 µm.

regard, it is important to keep in mind that transcription during early embryogenesis is stochastic, as only a subset of nuclei express the same gene at any given time (*Fukaya et al., 2016*; *Muerdter and Stark, 2016*; *Zoller et al., 2018*). Consequently, we used embryo counts in place of individual pole cell counts to compare between different samples, which likely underrepresents the frequency of the observed ectopic transcription events in PBs/PGCs. Nonetheless, as WT PBs or PGCs never display *sis-b* transcripts, our data show that Gcl is required to repress the transcription of XCEs in PBs and PGCs.

### *Sxl* RNA is detected in *gcl* PBs and PGCs

We next used smFISH to determine if the transcriptional target of the XCEs, the *Sxl-Pe* promoter, is active in *gcl* PBs and/or PGCs. In the soma, we found that the pattern of *Sxl-Pe* activity was

indistinguishable between WT and *gcl* embryos, as *Sxl-Pe* transcripts are not detected prior to nuclear migration, nor are they observed in NC 10 somatic nuclei. In approximately half of the embryos, *Sxl-Pe* transcripts are observed in somatic nuclei from NC 11 until NC 14. Moreover, in these embryos, two nuclear dots of hybridization are detected in most nuclei, indicating that they are female (*Erickson and Quintero, 2007*; *Keyes et al., 1992*). In the remaining *gcl* and WT NC 11–14 embryos, *Sxl-Pe* transcripts are not observed in somatic nuclei, indicating that these embryos are male.

While the pattern of *Sxl-Pe* activity in the soma of *gcl* embryos is the same as WT, this is not true in the germline. As shown in *Figure 1d*, *Sxl-Pe* transcripts can be detected in PBs and PGCs in 42% of *gcl* embryos (n = 31 embryos, p=0.001593), while transcripts are not observed in WT PBs or PGCs (*Figure 1c*, n = 18 embryos). It is notable that the *Sxl-Pe* promoter remains active after the PBs cellularize, and nascent *Sxl-Pe* transcripts can be detected in PGC nuclei of *gcl* embryos, while they are never observed in the WT PGCs. In *gcl* embryos, *Sxl-Pe* transcripts are found not only in female PGCs, as evidenced by *Sxl-Pe* expression in somatic nuclei, but also in male *gcl* PGCs, which lack somatic *Sxl-Pe*. In the NC 11–14 embryos examined, the frequency of female *gcl* embryos expressing *Sxl-Pe* transcripts in their PGCs is somewhat higher than that of male *gcl* embryos (*Table 1*). Two factors could contribute to this bias. First, *Sxl-Pe* promoter activity is turned on by XCEs (Sis-A, Sis-B, Runt) in a dose-dependent manner, and these XCEs are also *gcl* targets. Second, there are two copies of the *Sxl* gene in females, which could increase the probability that it will be active in *gcl* mutants.

To determine if the *Sxl-Pe* mRNAs detected in *gcl* PBs and PGCs are properly processed, exported, and translated, we probed WT and *gcl* embryos with Sxl antibodies. As *Sxl-Pe* is not activated in WT female embryos until NC 11, Sxl protein is only readily detectable in somatic nuclei during NC 13/14. It is normally absent in the somatic nuclei of male embryos and in the PGCs of both sexes. While the pattern of Sxl protein accumulation in the soma of *gcl* embryos is the same as WT, this is not true in PGCs. Sxl protein can be detected in the PGCs of *gcl* embryos (*Figure 2A–D*; WT control: n = 40; 2/40 Sxl positive PGC nuclei as opposed to *gcl*: n = 36; 16/36 Sxl positive, p=7.621e-05). These data indicate that the *Sxl-Pe* promoter is normally repressed by Gcl in PBs and newly formed PGCs. While the failure to turn off the ongoing transcription of *sis-b*, *sis-a* (and possibly *runt*) likely contributes to the activation of *Sxl-Pe* in *gcl* PBs and PGCs, the fact that activation of the promoter is earlier than normal and is subsequently observed in both female and male PGCs suggests that XCE activity may not be the only contributing factor.

## Ectopic expression of *gcl* represses *Sxl*

The experiments described above indicate that *gcl* is required to keep *Sxl* off in PBs and PGCs. We wondered whether *gcl* is sufficient to downregulate *Sxl* expression independent of other maternally derived components of the pole plasm, like *nos*, that are known to be required to keep the *Sxl* gene off. To address this question, we took advantage of a transgene in which the *gcl* mRNA protein coding sequence is fused to the *bicoid* (*bcd*) 3'UTR (*Jongens et al., 1994*; *Leatherman et al., 2002*). Using this transgene, *Leatherman et al., 2002* found that expression of Gcl at the anterior of the embryo induced a local reduction in the expression of *sis-b*, *sis-a*, as well as terminal patterning genes such as *tailless* and *huckebein*. Nuclear accumulation of Sxl protein is uniform across the WT control female embryo, including the anterior (n = 12), while male embryos are completely devoid of Sxl (n = 15). We found that Sxl protein accumulation was diminished in nuclei at the anterior of *gcl-bcd-3'UTR* female embryos. While reduction in Sxl was observed in all female embryos, it was readily

**Table 1.** PGC transcription in *gcl* embryos shows a slight, but not significant, sex bias.
Significance for sex ratios of embryos showing transcription in PBs and PGCs was determined using Fisher's exact test; p-values are displayed in the right column.

| | No PGC transcription | | PGC transcription | | p-value |
|---|---|---|---|---|---|
| | Male | Female | Male | Female | |
| WT | 14 | 18 | 0 | 0 | 1 |
| *gcl* | 12 | 15 | 5 | 14 | 0.235205 |

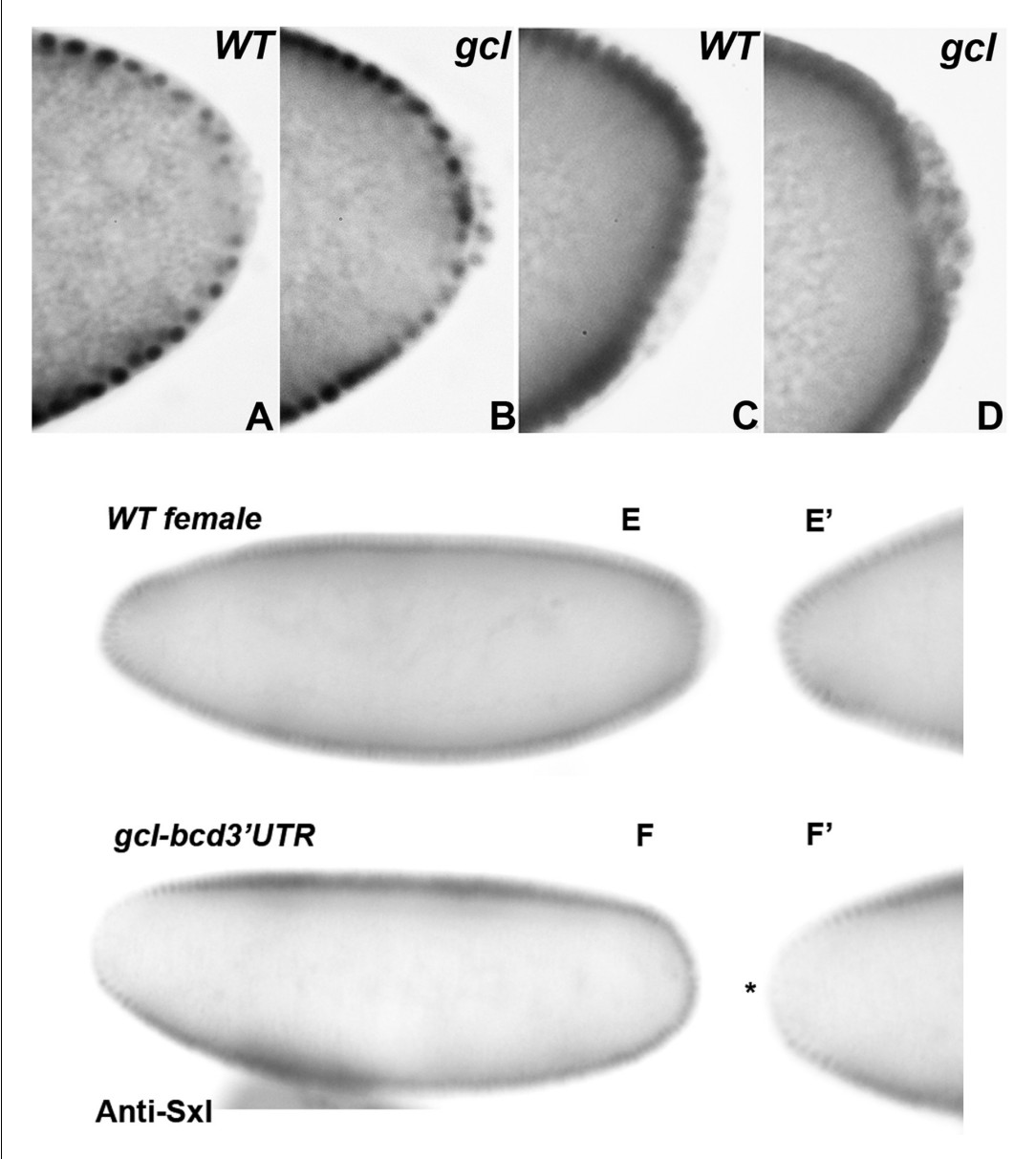

**Figure 2.** Gcl represses Sxl expression in the early embryonic pole cells, and ectopic expression of Gcl is sufficient to repress *Sxl* in somatic nuclei. 0–4 hr old paraformaldehyde-fixed embryos from mothers of indicated genotype were stained with anti-Sxl antibody to assess whether Sxl expression is upregulated in *gcl* PGCs (A–D). Posterior of the embryos are oriented to the right in all images. Panels **A** and **B**: early syncytial blastoderm stage embryos. Sxl protein is absent in the pole cells from the control (wild-type [WT]) embryo (**A**) whereas some of the *gcl* mutant pole cells show presence of Sxl (**B**). Panels **C** and **D**: Syncytial blastoderm stage female embryos from mothers of the indicated genotype were stained using anti-Sxl antibody. Similar to pole buds, only *gcl* mutant pole cells show Sxl protein (**D**) as opposed to the control (**C**). Panels **E**– **F'**: To determine whether Gcl is sufficient to repress Sxl expression on its own, embryos derived from females carrying *gcl-bcd* 3'UTR transgene (**F**) were stained using anti-Sxl antibodies. WT embryos were used as a control (**E**). The *gcl-bcd* 3'UTR transgene consists of genomic sequences of the *gcl* coding region fused to the 3'UTR of the anterior determinant *bcd,* resulting in ectopic localization of *gcl* mRNA to the anterior pole. Anterior poles are oriented to the left in each image. Images on the right in the panels **E'** and **F'** show just the anterior pole from the same embryos. While Sxl-specific signal is strong and uniform in the control embryo, selective reduction in Sxl in the anterior is readily seen in the *gcl-bcd* 3'UTR embryo (marked with an asterisk).

discernible in 9/13 embryos; p=2.23e-04. By contrast, Sxl was absent in *gcl-bcd-3'UTR* male embryos as in the case of control (n = 15) (*Figure 2E and F*). This localized disruption of Sxl expression is coincident with the anterior expression of Gcl protein in the *gcl-bcd-3'UTR* embryos, indicating that Gcl alone is sufficient to repress Sxl.

## Premature expression of *Sxl* in the PGCs leads to germ cell loss and defective germ cell migration

Since our findings indicate that *Sxl* is inappropriately expressed in *gcl* PBs and newly formed PGCs, an important question is whether precocious expression of *Sxl* has detrimental effects on PGC development. To test this possibility, we ectopically expressed *Sxl* in early embryos. We mated *maternal-tubulin-GAL4* (referred to as *mat-GAL4*) virgin females with males carrying a *UAS-Sxl* transgene. The maternally deposited GAL4 was able to drive the zygotic expression of Sxl protein in early female and male embryos independent of its normal regulation. We compared the total number of PGCs in late syncytial and early cellular blastoderm (stage 4/5) *mat-GAL4/UAS-Sxl* with *mat-GAL4* embryos. In WT, there are typically about 25 PGCs in stage 4/5 embryos. This number is reduced nearly two-fold in *mat-GAL4/UAS-Sxl* embryos (**Figure 3**). A reduction of PGCs was also observed when we mated virgins carrying the germ cell-specific *nosGAL4-VP16* promoter to *UAS-Sxl* males to drive expression in the germline (6.5 PGCs per gonad in *nosGAL4/UAS-Sxl* embryos, n = 15, compared to 10 PGCs per gonad in *nosGAL4/+* control, n = 12 embryos). Further, overexpression of *Sxl* in the germline impaired PGC migration. **Figure 4** shows PGC migration defects in *nosGAL4-VP16/UAS-Sxl* embryos (3/21 *UAS Sxl/+* control embryos showed >5 mispositioned PGCs as opposed to 9/17 *nosGAL4-VP16/UAS-Sxl* embryos; p=0.04). Taken together, these findings demonstrate that precocious expression of Sxl protein has deleterious effects on PGC development and behavior during early embryogenesis.

## Simultaneous removal of *gcl* and *Sxl* ameliorates the *gcl* phenotype

The finding that premature ectopic expression of Sxl protein has adverse effects on PGC development supports the idea that one critical function of *gcl* is repressing *Sxl-Pe*. If this is correct, then compromising *Sxl* activity in the early embryo should mitigate the PGC defects seen in *gcl* embryos. For this purpose, we generated *gcl* embryos that also carry a small deficiency, $Sxl^{7BO}$, which deletes the *Sxl* gene. In this experiment, we mated $Sxl^{7BO}$/*Bin*; *gcl/gcl* mothers to $Sxl^{7BO}$/Y fathers, and 0–12 hr old progeny were probed with Sxl and Vasa antibodies. While all of the progeny from this cross lack maternally derived Gcl protein, only half of the progeny lack the *Sxl* gene. For female embryos, one half would be $Sxl^{7BO}$/*Bin*, while the other half would be $Sxl^{7BO}$/$Sxl^{7BO}$. The former ($Sxl^{7BO}$/*Bin*) have a functional *Sxl* gene, and, since they are females, they will express Sxl protein in the soma, which can be detected with Sxl antibody. The latter ($Sxl^{7BO}$/$Sxl^{7BO}$) do not have a functional *Sxl* gene and would not express Sxl protein even though they are female. There are also two classes of male embryos. One half would be *Bin/Y*, while the other half would be $Sxl^{7BO}$/Y. The former (*Bin/Y*) has a functional *Sxl* gene, but since they are males (with a single X chromosome), *Bin/Y* embryos would not express Sxl protein. The latter, $Sxl^{7BO}$/Y lacks a functional *Sxl* gene and would also not express Sxl protein.

To identify the different classes of embryos, we stained with Sxl antibody. Using this approach, we can unambiguously identify the genotype of the $Sxl^{7BO}$/*Bin*, as they express Sxl protein throughout the soma. One quarter of the embryos fall into this class. The remaining three quarters of the embryos do not express Sxl protein, and we cannot unambiguously identify their genotype or sex. However, we know that one third of the embryos that do not stain with Sxl antibody are *Bin/Y* males and have thus a functional *Sxl* gene. The remaining embryos (two thirds of the embryos that do not stain with Sxl antibody, or one half of the total embryos in the collection) are either $Sxl^{7BO}$/$Sxl^{7BO}$ females or $Sxl^{7BO}$/Y males, and, in both cases, they lack a functional *Sxl* gene.

If removal of *Sxl* ameliorates the *gcl* defects in PGC formation, then we should observe an increase in the number of PGCs in only one half of the embryos from this cross. Moreover, this increase should be observed in the embryos that do not stain with Sxl antibody. However, within the group of embryos that do not stain with Sxl antibody, only two thirds should show an increased number of PGCs. All of these expectations are met. The graph in **Figure 5** shows that the average number of PGCs in $Sxl^+$ *gcl* ($Sxl^{7BO}$/*Bin*) (mean ~3, n = 14) (female) embryos is not too different from that in *gcl* embryos (mean ~2, n = 24) that are WT for *Sxl*. In the class of embryos that lack Sxl protein, there are two unequal groups, as expected. In one group, which corresponds to about one third of the unstained embryos, the mean number of PGCs is 3.5. This group matches closely with the number of PGCs in $Sxl^+$ ($Sxl^{7BO}$/*Bin*) females and thus presumed to be $Sxl^+$ (*Bin/Y*) males. In the other group, which corresponds to about two thirds of the unstained embryos (or half the total number of

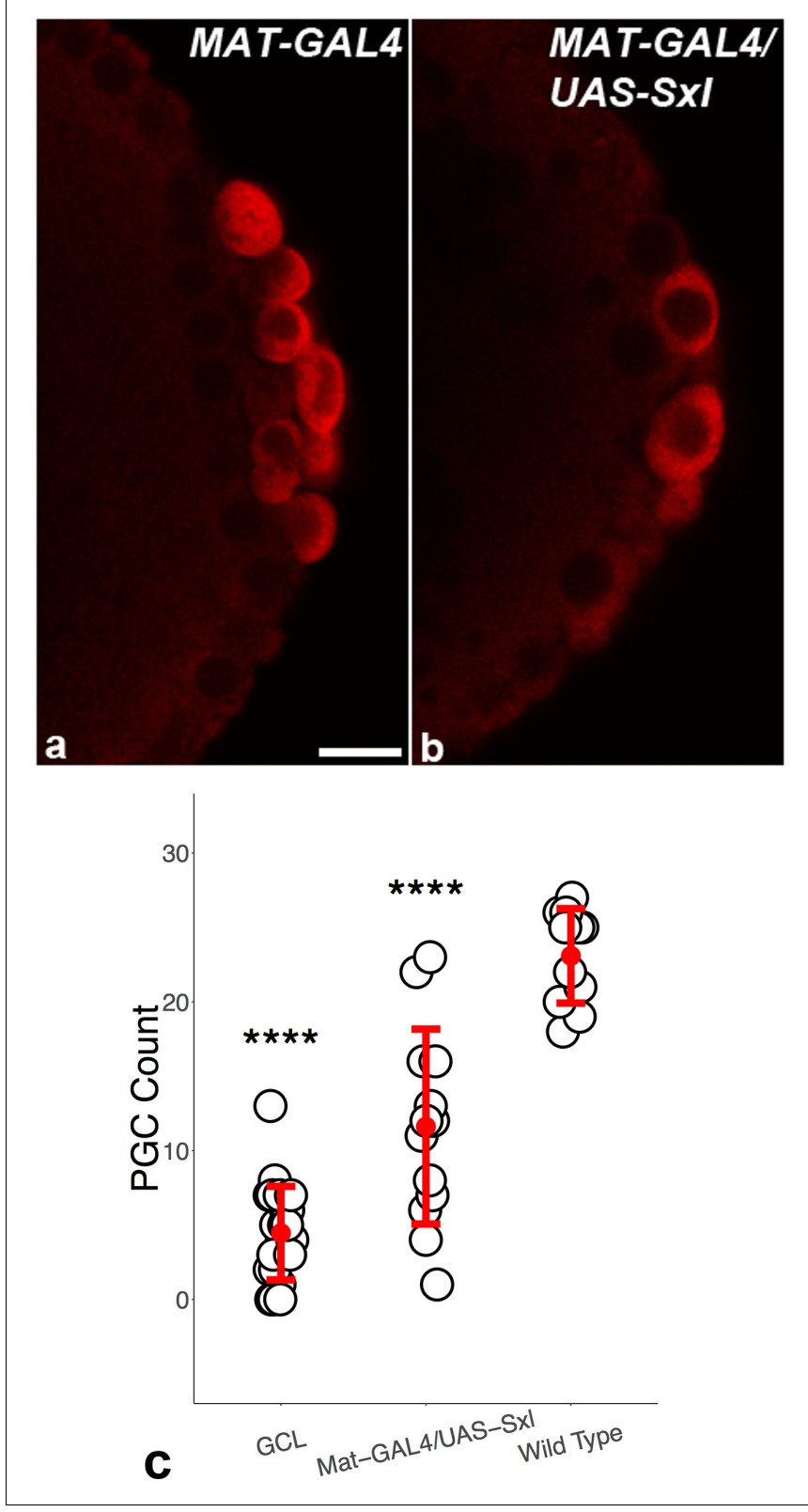

**Figure 3.** Precocious expression of *Sxl* results in reduction in total number of primordial germ cells (PGCs). Embryos of indicated genotypes were stained for pole cell marker Vasa (panels **a** and **b**; imaged in red) to discern the effects of precocious *Sxl* activity on the early PGCs. UAS-*Sxl* transgene males were mated with females carrying *maternal-tubulin-GAL4* driver (panel **b**) to assess if precocious *Sxl* expression adversely influences early

*Figure 3 continued on next page*

*Figure 3 continued*

PGCs. *mat-GAL4* (panel **a**) and *gcl* (not shown) embryos served as positive and negative controls, respectively. (**c**) Quantitation of PGC counts in different genetic backgrounds. The number of pole cells in embryos from mothers of indicated genotypes were counted and compared. Bars represent the mean ± S.D. (n = 23 for *gcl*, n = 14 for *mat-GAL4/UAS-Sxl*, n = 12 for *mat-GAL4*). ****p<0.0001 for *gcl* and *mat-GAL4/UAS-Sxl* compared to wild type (WT). Note that *p>0.01 for *gcl* compared to *mat-GAL4/UAS-Sxl* (not indicated in the graph). The online version of this article includes the following source data for figure 3:

**Source data 1.** Primordial germ cell (PGC) counts.

embryos), the mean number of PGCs is 12. Embryos in this group are presumably *Sxl⁻* males and females. The combined average of the PGC count for all of the embryos that do not stain with Sxl antibody is ~8.5, which is also significantly higher than *gcl* (~2; see **Figure 5** legend for details). These findings indicate that removing the *Sxl* gene ameliorates the effects of the *gcl* mutation on PGCs.

## RNAi knockdown of *Sxl* also ameliorates the PGC formation defects in *gcl* embryos

To confirm that ectopic activation of *Sxl-Pe* in *gcl* mutants has deleterious effects on PGCs, we also used *RNAi* to knockdown expression of Sxl protein. *gcl* mothers carrying a *mat-GAL4* driver were mated to males carrying *UAS-Sxl-RNAi* transgene, and the embryos derived from this cross were stained with anti-Vasa antibodies to visualize PGCs. **Figure 6** shows that *RNAi* knockdown of *Sxl* (*Sxl^RNAi*) partially suppresses the effects of the *gcl* mutation on PGCs. While all the embryos in this

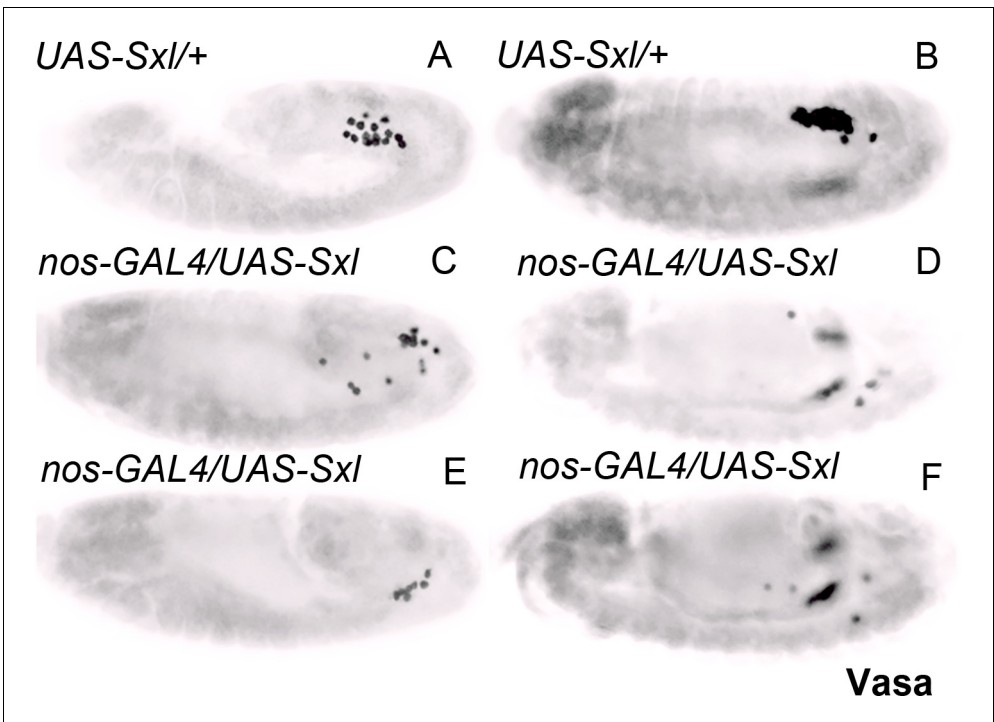

**Figure 4.** Germ cell-specific expression of Sxl leads to germ cell migration defects during mid-embryogenesis. Embryos from mothers of the indicated genotypes were stained for the germ cell marker Vasa. *UAS-Sxl* transgene males were mated with virgin females carrying the germline-specific driver *nos-GAL4-VP16* to assess if precocious *Sxl* expression can influence PGC migration and survival (panels **C–F**). Embryos at stage 12 (**A, C, E**) and stage 13 (**B, D, F**) are shown as germ cell behavior defects become apparent from stage 12 onwards. *UAS-Sxl/+* embryos served as control (**A and B**). Readily detectable germ cell migration defects were seen in the experimental embryos as opposed to the control. 3/21 *UAS Sxl/+* control embryos showed >5 mispositioned PGCs as opposed to 9/17 *nosGAL4-VP16/UAS-Sxl* embryos; p=0.04 (significance determined using Welch's two sample t-test).

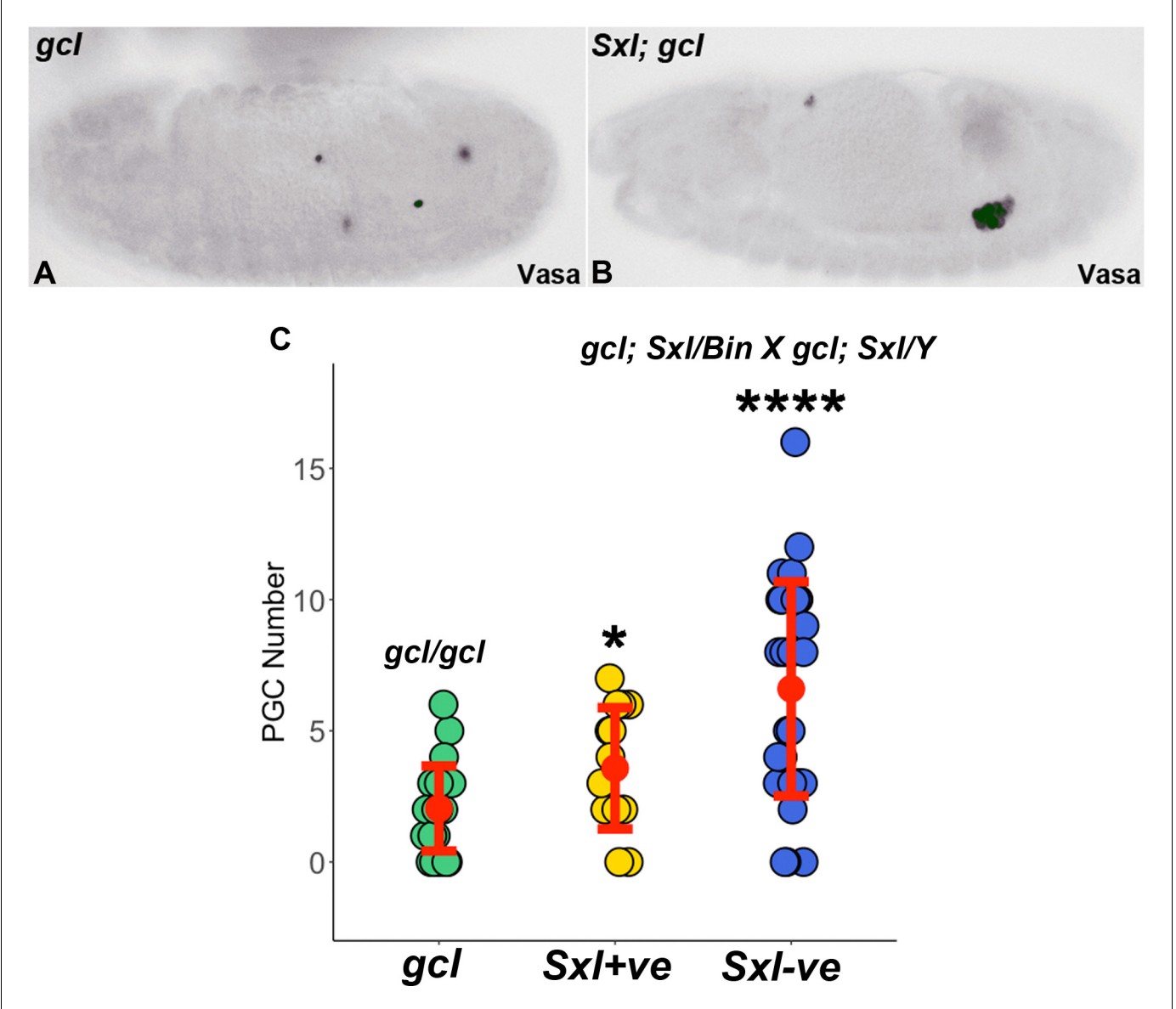

**Figure 5.** Simultaneous removal of *gcl* and *Sxl* mitigates the *gcl* phenotype. (**A-B**) 0–12 hr old embryos (from the cross *7BO/Y;gcl/gcl x7BO/Bin;gcl/gcl*) were stained using anti-Sxl antibody and for the germline marker Vasa. 7BO is a small deficiency chromosome that specifically deletes the *Sxl* gene. Embryos that stained positive for Sxl were disregarded (n = 14) since only embryos lacking *Sxl* and *gcl* are relevant in this experiment. Male embryos of genotype *Bin/Y; gcl/gcl* (**A**) are compared with embryos believed to be of genotype *7BO/7BO; gcl/gcl* or *7BO/Y; gcl/gcl*(**B**). The number of pole cells in embryos from mothers of indicated genotypes were counted and plotted (**C**). Bars represent the mean ± SD (n = 23 for *7BO/7BO; gcl/gcl*, n = 19 for *7BO/Y; gcl/gcl*, n = 26 for *Bin/Y; gcl/gcl*). ****p<0.0001 for *7BO/7BO; gcl/gcl* and *7BO/Y; gcl/gcl* compared to *Bin/Y; gcl/gcl*. p=0.03 for *7BO/7BO; gcl/gcl* compared to *7BO/Y; gcl/gcl*. Significance was determined using Welch's two sample t-test.

The online version of this article includes the following source data for figure 5:

**Source data 1.** Primordial germ cell (PGC) counts.

experiment were of identical genotype, they fell into two classes: one in which the number of PGCs in syncytial/early cellular blastoderm embryos is nearly WT and another that had few PGCs.

A plausible explanation for this bimodal distribution is that the efficiency of rescue reflects sex-specific differences in the dose of X-linked sex-determination genes. Females have two copies of not only *Sxl* but also the XCEs responsible for activating *Sxl-Pe*, whereas males have only a single copy of these genes. Consistent with gene dose being relevant, there is a modest female-specific bias in the frequency in which we detect *Sxl-Pe* transcripts in *gcl* PBs/PGCs (*Table 1*). To test this directly,

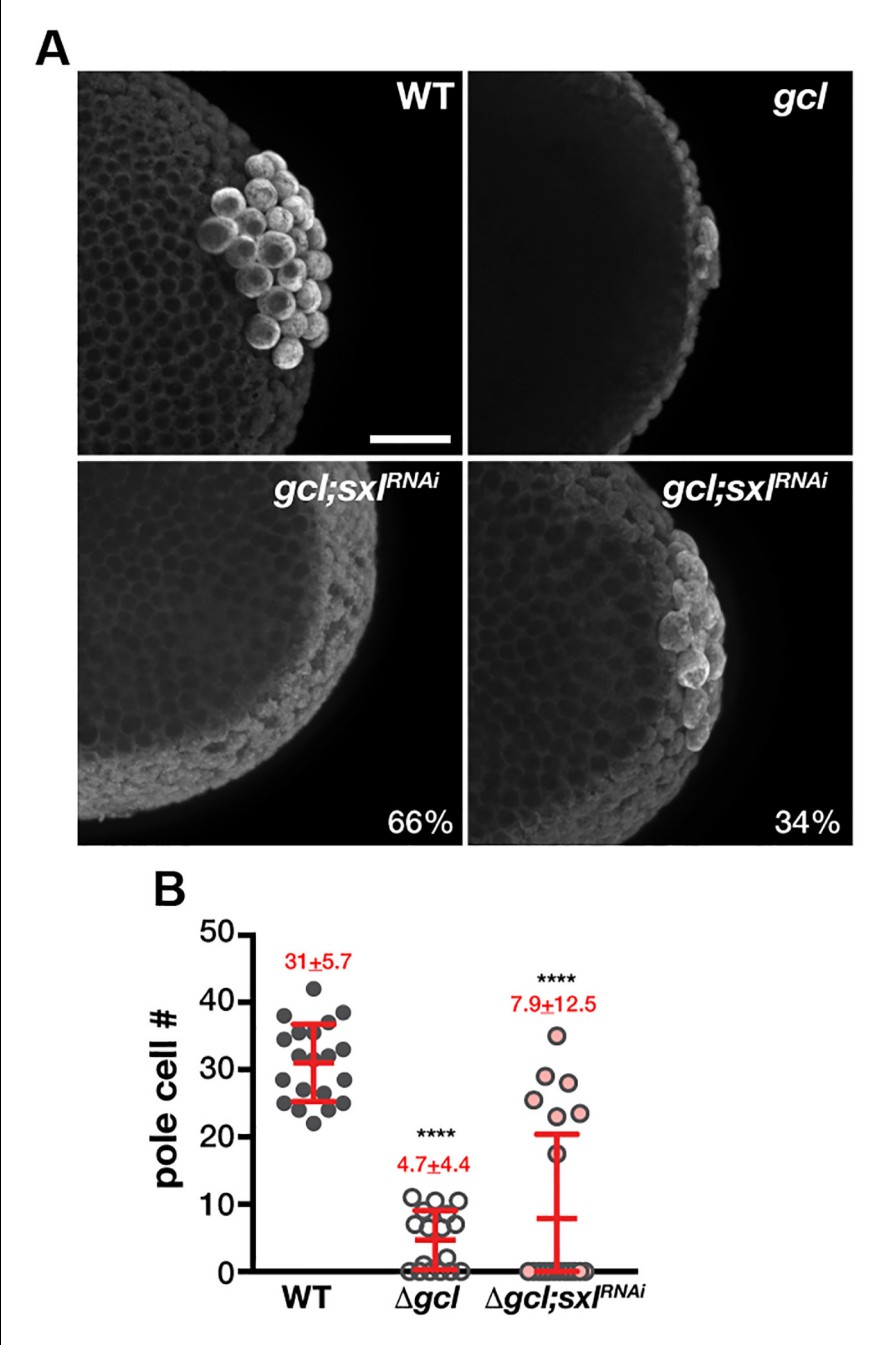

**Figure 6.** Knockdown of *Sxl* partially suppresses germ cell loss of *gcl* embryos. *gcl;mat-GAL4* virgin females were mated with males carrying *UAS-Sxl RNAi*. Embryos derived from this cross were stained with anti-Vasa antibody to visualize PGCs (**A**). Scale bar represents 20 μm. Total number of PGCs were counted for each embryo from different genotypes, and a Mann–Whitney U-test was employed to analyze significant differences between wild type (WT), *gcl*, and *gcl;Sxl^RNAi^* (**B**). In 66% of *gcl;Sxl^RNAi^* embryos, few or no pole cells are observed, comparable to *gcl*. However, in 34% of *gcl;Sxl^RNAi^* embryos, germ cell count is substantially elevated, indicating partial rescue of the *gcl* phenotype.

The online version of this article includes the following source data for figure 6:

**Source data 1.** Primordial germ cell (PGC) counts.

we determined the sex of the *gcl* and control embryos using smFISH with *sis-b* and *Sxl* probes. At the syncytial blastoderm stage somatic nuclei in female embryos have two dots of hybridization for both *sis-b* and *Sxl*. By contrast, male embryos have one dot of hybridization for *sis-b* and no signal for *Sxl* (*Figure 7*). When we stained embryos derived from the experimental cross, we observed that all embryos showing an increase in PGC formation were females (*Sxl*⁺ and two dots of *sis-b* signal) (*Table 2*, n = 59, p=0.002456).

## Ectopic transcription is attenuated in *gcl;tsl* PGCs

In their studies showing that Gcl targets the terminal pathway receptor Torso for proteolysis, *Pae et al., 2017* found that mutations in the *torso-like* (*tsl*) ligand modifier or *RNAi* knockdown of *torso* also suppressed the PGC defects in *gcl* embryos. We confirmed that simultaneous removal of maternal *tsl* and *gcl* resulted in a substantial rescue of the PGC formation defects in *gcl* embryos (*Pae et al., 2017*). *Figure 8* and *Table 3* show that *gcl;tsl* embryos display a significant increase in the number of PGCs as compared to *gcl* embryos, and that the rescue is highly penetrant (p<2e-16, *Figure 8D*). (Note also that the rescue is more substantial than that observed in the *Sxl* experiments.)

*Leatherman et al., 2002* found that *gcl* was required for turning off somatic gene transcription in PBs/PGCs, and they suggested that one of the critical functions of *gcl* in PGC formation is the silencing of transcription. In addition to confirming that *gcl* is required to turn off transcription in PBs/PGCs, we also identified an important target for *gcl* mediated repression, the *Sxl* establishment promoter, *Sxl-Pe*. Taken together with the fact that removal of *tsl* gives nearly complete rescue of the PGC formation defects in *gcl*, these observations would imply that *gcl* must target Torso for degradation (at least in part) in order to block the terminal pathway from promoting the transcriptional activity of somatic genes (including activation of *Sxl-Pe*). If this prediction is correct, then the misexpression of *Sxl-Pe* and other genes should not be observed in embryos from *gcl;tsl* mothers where the PGC formation defects are rescued. To test this prediction, we performed smFISH on *gcl;tsl* embryos using *Sxl* and *sis-b* probes along with *gcl* and WT embryos as positive and negative controls, respectively. *Table 3* shows that removal of *tsl* restores transcriptional quiescence in the PBs/PGCs of *gcl;tsl* embryos (*Figure 8* and *Table 3*, p=1e-06 and 1 for WT compared to *gcl* and *gcl;tsl*, respectively, by Fisher's exact test). Taken together, these data confirm that inactivation of the

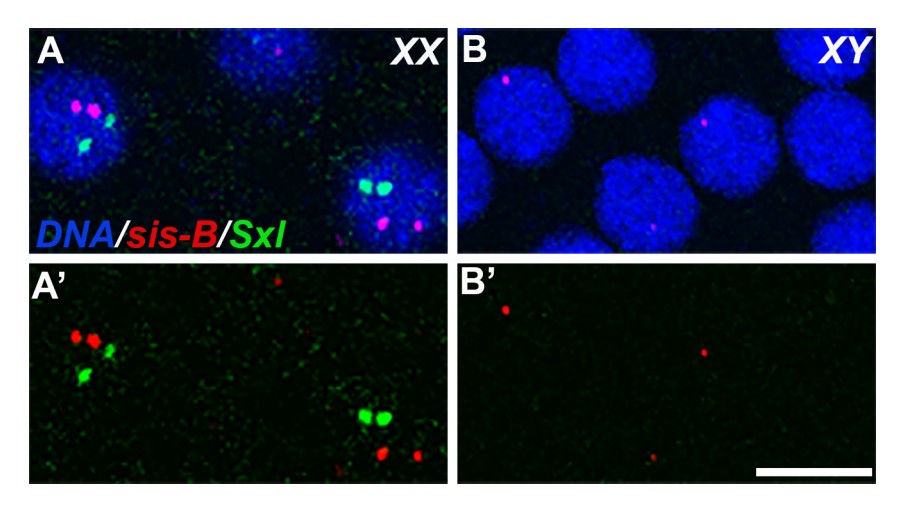

**Figure 7.** Sexing embryos based on transcription puncta from X-chromosomes. 0–3 hr old wild-type (WT) embryos were probed for *Sxl* (green) and *sis-b* (red) transcription using smFISH, and these embryos were co-stained with Hoescht to visualize DNA. (**A**) Embryos with two X-chromosomes (females) show two transcription puncta for both *sis-b* and *Sxl*, corresponding to expression from each X. (**B**) XY embryos (males) transcribe *sis-b* from the only X chromosome and fail to activate expression of *Sxl*. (**A and B**) show merge; (**A' and B'**) show smFISH signals. A representative section of somatic nuclei is shown in each panel. Scale bar represents 10 μm.

**Table 2.** Rescue of PGC numbers in *gcl;Sxl^RNAi* embryos only occurs in female embryos.

|  | Male | Female |
| --- | --- | --- |
| No rescue | 20 | 26 |
| Rescue | 0 | 13 |

terminal signaling pathway by Gcl is critical for silencing transcription in PBs and PGCs and that this silencing function plays an important role in PGC formation.

## A degradation-resistant form of Torso also activates transcription in PGCs

Our finding that the survival of *gcl;tsl* PGCs is accompanied by the reestablishment of transcriptional silencing provides strong support for the idea that *gcl* targets Torso for degradation to block terminal signaling dependent transcription. A prediction of this model is that transcription of *gcl* targets should be ectopically activated in PBs/PGCs when Gcl-dependent proteolysis of Torso is blocked. To test this prediction, we took advantage of a mutant transgene version of *torso*, *torso^Deg*, generated by *Pae et al., 2017*, that lacks the Gcl interaction domain and is thus resistant to Gcl-dependent proteolysis. Embryos from females carrying both *mat-GAL4* and *UAS-torso^Deg* were probed for *sis-b* and *Sxl-Pe* promoter activity. *Figure 9* shows that both *sis-b* and *Sxl-Pe* transcripts are inappropriately expressed in the PBs and PGCs of *torso^Deg* embryos, with frequencies less than those

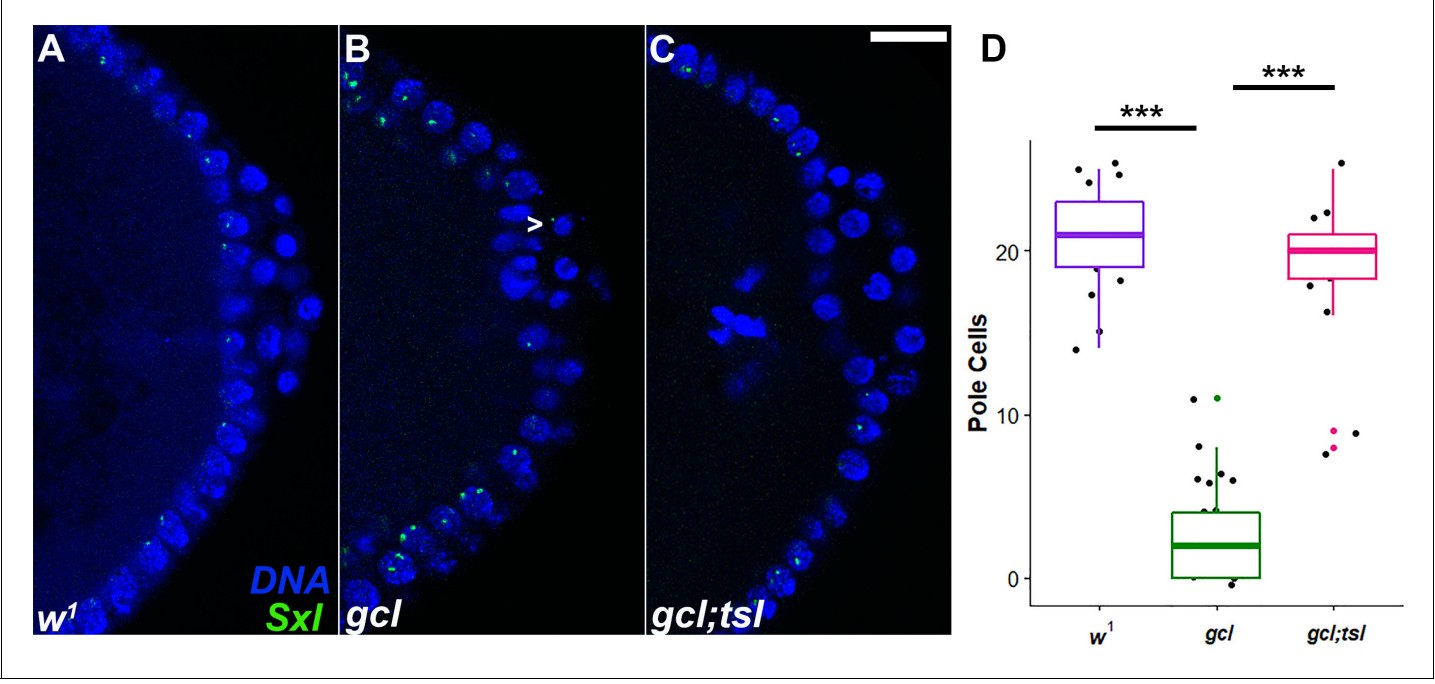

**Figure 8.** Rescue of PGCs in *gcl;tsl* embryos. smFISH using *Sxl* probes was performed to assess the status of transcription in PBs of wild-type (WT) (A), *gcl* (B), and *gcl;tsl* (C) 0–3 hr old embryos. Posterior poles of representative blastoderm embryos are shown with *Sxl* RNA visualized in green and Hoescht DNA dye in blue. While 0% of control embryos display *Sxl* transcription in PBs, transcription of *Sxl* is detected in 67% buds of *gcl* embryos (indicated with a carrot in the representative embryo). In *gcl;tsl* embryos, however, 0% display any ectopic transcription (*Table 3*). n = 28, 23, and 24 for WT, *gcl*, and *gcl;tsl* embryos, respectively; by Fisher's exact test, p=1e-06 and 1 for WT compared to *gcl* and *gcl;tsl*, respectively, and p=2e-06 for *gcl* compared to *gcl;tsl*. Scale bar represents 10 μm. (D) Pole cell counts from WT, *gcl*, and *gcl;tsl* embryos were counted using anti-Vasa staining (n = 17, 25, and 18, respectively). ***p<0.001 for the compared genotypes shown. Significance was determined using a one-Way ANOVA (p=0) with pairwise t-test comparisons (p=0 for WT vs. *gcl*, p=0.14 for WT vs. *gcl;tsl*, p=0 for *gcl* vs. *gcl;tsl*). These data suggest that rescue of the *gcl* PGC numbers is highly penetrant in *gcl;tsl* embryos.

The online version of this article includes the following source data for figure 8:

**Source data 1.** Primordial germ cell (PGC) counts.

**Table 3.** Transcription status in PBs and PGCs of wild-type (WT), *gcl*, and *gcl;tsl* embryos (assessed using smFISH for *sis-b* and *Sxl*).
Significance for proportions of embryos showing transcription in PBs and PGCs was determined using Fisher's exact test; p-values are displayed in the right column.

| Genotype | No transcription | Transcription | p-value |
|---|---|---|---|
| WT | 28 | 0 | |
| *gcl* | 9 | 14 | 1.00e-06 |
| *gcl;tsl* | 24 | 0 | 1 |

observed in *gcl* embryos but significantly more than control embryos (27% of $torso^{Deg}$ embryos express *sis-b* (n = 16, p=0.043382) and 28% of $torso^{Deg}$ embryos express *Sxl* (n = 25, p=0.030307)). Thus, ectopic upregulation of *Sxl* and *sis-b* transcription observed in *gcl* pole cells is recapitulated in $torso^{Deg}$ embryos.

Taken together with the data reported by *Leatherman et al., 2002*, our results indicate that ectopic expression of Gcl at the anterior of the embryo downregulates transcription of multiple genes. If the relevant target for *gcl* in *gcl-bcd-3'UTR* embryos at the anterior is the Torso receptor, then we would predict that $torso^{Deg}$ should impact transcription not only in the germline, but also in the soma. Since the X-chromosome counting system, which regulates *Sxl-Pe* activity, is (at least partially) overridden in $torso^{Deg}$ PBs and PGCs, it seemed possible that it might also be overridden in the soma. To test this possibility, we examined *Sxl-Pe* expression in the soma of $torso^{Deg}$ embryos. In WT females, *Sxl-Pe* transcripts can be detected in virtually all somatic nuclei, and two dots of hybridization are typically observed (*Figure 7*). In males, *Sxl-Pe* is off and their somatic nuclei are completely devoid of the signal. While female $torso^{Deg}$ embryos resemble WT, we observed scattered nuclei in which *Sxl-Pe* is active in 43% of $torso^{Deg}$ male embryos (*Figure 10C*, n = 14, p=0.023871). This finding is also consistent with earlier studies in which we found that a constitutively active form of the Torso receptor, RL3, turns on *Sxl-Pe* in males (*Deshpande, 2004*).

## Does Gcl target a non-canonical Torso-dependent signaling pathway?

In the canonical terminal pathway, binding of the Tsl ligand to Torso activates a MAP kinase cascade that ultimately results in the phosphorylation and subsequent degradation of the transcriptional repressor Capicua by the ERK kinase (*de las Heras and Casanova, 2006*; *Grimm et al., 2012*). Degradation of Capicua, in turn, results in the transcription of terminal patterning genes, such as *tailless*. Surprisingly, however, *Pae et al., 2017* found that unlike *RNAi* knockdowns of the *torso* receptor, *RNAi* knockdown of two terminal pathway kinases, *dsor1* (MEK) and *rolled* (MAPK) that function downstream of Torso, failed to rescue the PGC defects of *gcl* embryos. From this finding, the authors concluded that Gcl-mediated degradation of the Torso receptor must disrupt the operation of a novel non-canonical Torso signaling pathway. To test the possibility that this non-canonical pathway might have a transcriptional output like the canonical transduction pathway, we used *mat-GAL4* to drive the expression of two activated versions of the MAPK/ERK kinase (MEK$^{E203K}$ and MEK$^{F53S}$ *Goyal et al., 2017*) in mothers and then assayed *Sxl-Pe* transcription in PBs and PGCs of their progeny. We found that maternal deposition of MEK$^{E203K}$ or MEK$^{F53S}$ could not activate *Sxl-Pe* transcription in PBs or PGCs (not shown, see Discussion). Nevertheless, we found that, as was observed for $torso^{Deg}$, *Sxl-Pe* expression is activated in male somatic nuclei by the GOF MEK proteins (*Figure 10D*, 46% of MEK$^{E203K}$ males showed patchy somatic *Sxl* expression, n = 13, p=0.019079). Taken together with our previous findings (*Deshpande, 2004*), this result would argue that the canonical Torso signaling pathway is capable of impacting *Sxl-Pe* promoter activity. In this context, it is also interesting to note that a key transcriptional target of the terminal signaling pathway, *tailless*, is not activated in *gcl* PBs or PGCs. This is also true for embryos expressing Torso$^{Deg}$ or either of the GOF MEK variants (not shown). Since *tailless* transcription is ectopically activated in *pgc* mutant PGCs, it would appear that the canonical terminal signaling pathway is not able to overcome the repressive effects of the Pgc protein in the case of *tailless*, even in a *gcl* background.

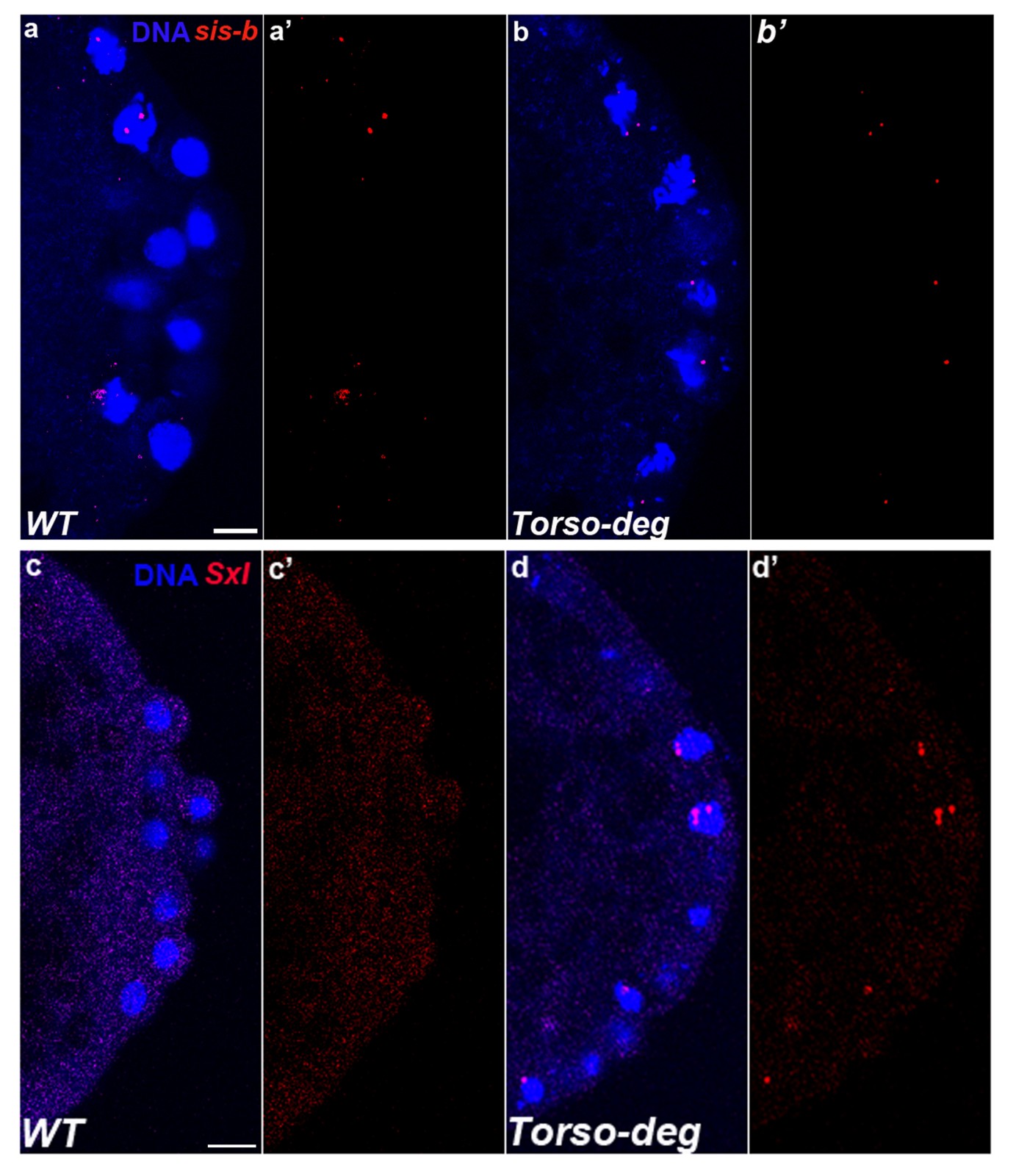

**Figure 9.** Transcriptional quiescence in pole cells is compromised in *torso*[Deg] embryos. smFISH using probes specific for *sis-b* or *Sxl* in 0–3 hr old embryos was performed to assess the status of transcription in *torso*[Deg] PBs. Posterior poles of representative pre-syncytial blastoderm embryos are shown with *sis-b* (a/b) or *Sxl* (c/d) RNA visualized in red and Hoescht DNA dye in blue. While 0% of control embryos display *sis-b* (a/a', n = 16) or *Sxl* (c/ c', n = 18) transcription in PBs, transcription of both *sis-b* (b/b') and *Sxl* (d/d') is detected in buds of *torso*[Deg] embryos. Note that transcription in wild-

*Figure 9 continued on next page*

*Figure 9 continued*

type (WT) embryos is only in somatic nuclei (**a**). We observed *sis-b* transcription in 27% (n = 15, p=0.043382) and *Sxl* transcription in 28% (n = 25, p=0.030307) of *torso^Deg^* embryos. Scale bar represents 10 µm.

## *torso^Deg^* disrupts the sequestration of germline determinants

One of the more striking phenotypes in *gcl* mutants is a failure to properly sequester protein and mRNA components of the pole plasm. In WT embryos, nuclei entering the posterior pole trigger the release of the pole plasm from the posterior cortex of the embryo by a centrosome/microtubule-dependent mechanism (*Lerit and Gavis, 2011*; *Raff and Glover, 1989*). Once released, the pole plasm constituents are distributed within the growing bud by a microtubule-dependent mechanism. However, spreading is restricted to the growing bud and the pole plasm components are ultimately incorporated into newly formed PGCs when the buds cellularize. In *gcl* embryos, nuclear entry also triggers the release of the pole plasm from the cortex; however, the pole plasm proteins and mRNAs are not retained in the newly formed PBs after they are released, but instead spread to the cytoplasmic territories of neighboring somatic nuclei along the cortex and also into the interior of the embryo (*Lerit et al., 2017*). The difference between WT and *gcl* in the localization of pole plasm constituents is shown for Vasa protein (*Figure 11*) and *gcl* (*Figure 12*), *pgc* (*Figure 13*), and *nos* (*Figure 14*) mRNAs. As shown in maximum intensity projections and the accompanying distribution graphs, Vasa and the three pole plasm mRNAs are sequestered in the PGCs of WT embryos. In contrast, in *gcl* embryos, Vasa protein, and *pgc* and *nos* mRNAs spread into the territories occupied by nearby somatic nuclei. As evident from the profiles of pole plasm distribution for individual embryos, the extent of spreading varies somewhat from embryo to embryo; however, retention of pole plasm constituents in PGCs is clearly disrupted in *gcl* embryos. In single sections, we also observe pole plasm constituents spreading into the interior of the embryo as well as along the posterior lateral cortex. We also detected no *gcl* mRNA in the *gcl* mutant, as expected (*Figure 12*).

Interestingly, as was the case for transcriptional activation, the effects of *torso^Deg^* on the sequestration of the pole plasm constituents are quite similar to those observed in *gcl* embryos. In early *torso^Deg^* embryos, pole plasm constituents appear to be localized correctly to the posterior pole (*Figure 15*). However, after the nuclei migrate to the surface of the embryo, the localization of pole plasm components is disrupted. Vasa protein (*Figure 11*) and *pgc* (*Figure 13*) and *nos* (*Figure 14*)

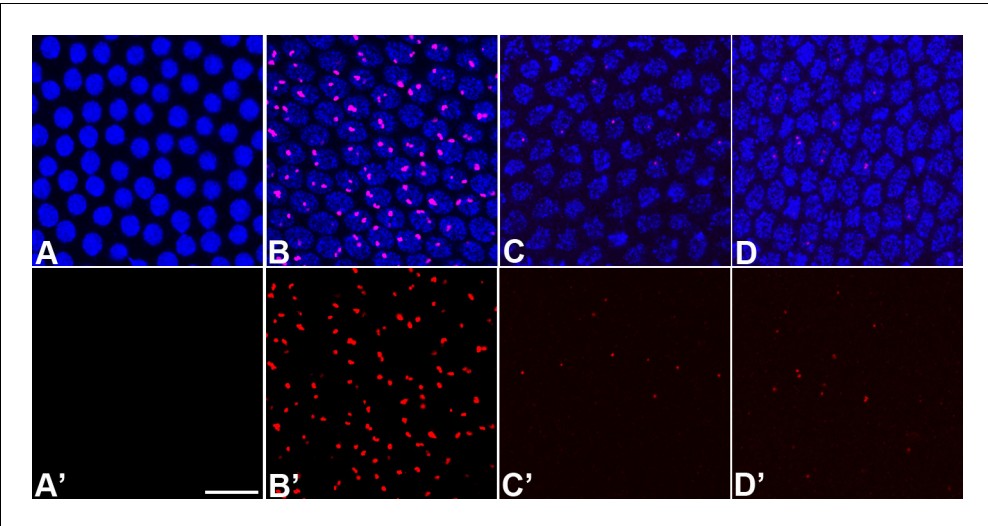

**Figure 10.** *Sxl* is expressed in the male soma in *torso^Deg^* and MEK GOF embryos. 0–3 hr old embryos were probed for somatic *Sxl* transcription using smFISH. While 0% of control male embryos display *Sxl* expression in the soma (**A and A'**, n = 10), all control females display uniform somatic *Sxl* expression (**B and B'**, n = 17). However, we observed sporadic somatic *Sxl* activation in 43% (n = 14, p=0.023871) of *torso^Deg^* (**C and C'**) and 46% (n = 13, p=0.019079) of *MEK^E203K^* (**D and D'**) male embryos. A representative section of somatic nuclei is shown in each panel (blue) with *Sxl* transcripts in red. Scale bar represents 10 µm.

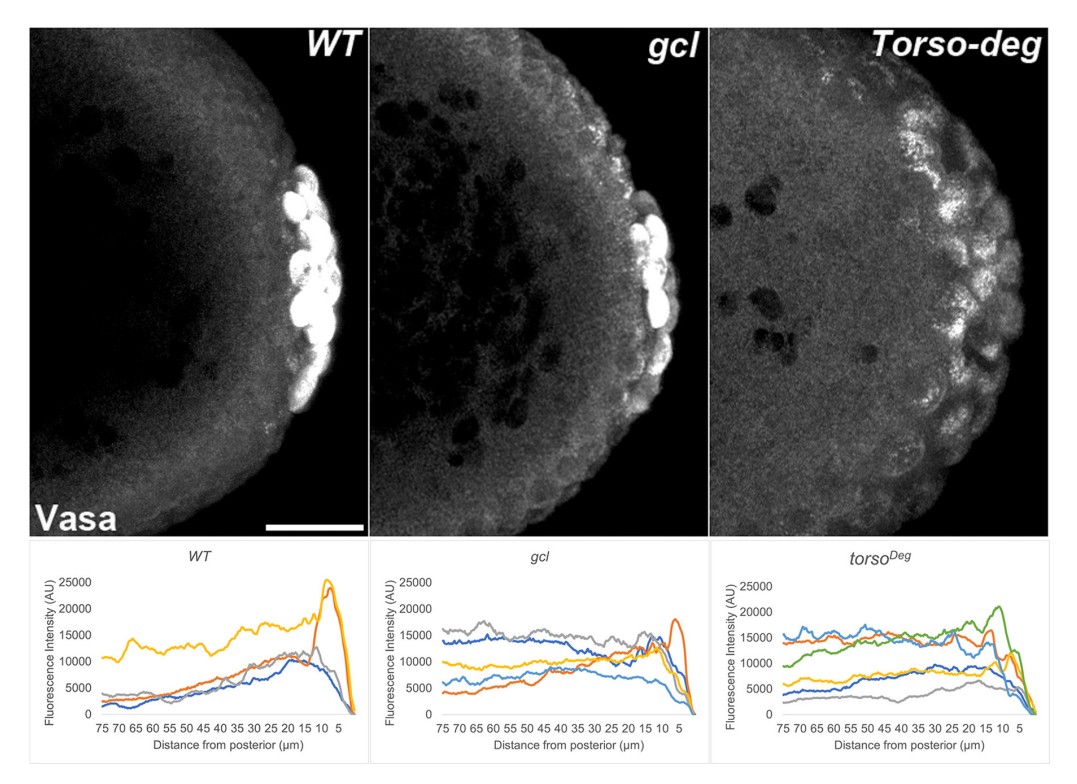

**Figure 11.** Vasa is mislocalized from the posterior in *gcl* and *torso^Deg* embryos. 0–3 hr old paraformaldehyde-fixed embryos collected from wild-type (WT), *gcl*, or *torso^Deg* mothers were stained with anti-Vasa antibody to assess whether pole plasm is properly localized in *gcl* and *torso^Deg* embryos. On top, images are representative maximum intensity projections of the posterior pole of each indicated genotype. Scale bar represents 10 μm. Below, plot profiles show mislocalization of pole plasm (visualized using Vasa) away from posterior cap in *gcl* and *torso^Deg* embryos (see Materials and methods for details of quantification). Each plot shows a representative experiment, with each line depicting pole plasm distribution of an individual embryo. n = 12, 13, and 13 for WT, *gcl*, and *torso^Deg*, respectively.

mRNAs spread into the territories of somatic nuclei located along the posterior lateral cortex of *torso^Deg* embryos. In addition, *gcl* mRNA (*Figure 12*) is not properly restricted in *torso^Deg* embryos, and like *pgc* and *nos* mRNAs, it is distributed along the lateral cortex. This finding is of special interest as it suggests the existence of an antagonistic relationship between *torso* and *gcl*. While *gcl* negatively regulates the Torso receptor by promoting its degradation, Torso activity likely controls the sequestration of pole plasm—including *gcl* mRNA—to the PBs and PGCs. Such a mechanism would avoid inappropriate exposure of the neighboring somatic nuclei to *gcl* RNA (and possibly protein), ultimately ensuring proper germline/soma distinction.

## Sequestration of germline determinants is disrupted by activated MEK

Although we found that ectopically expressed GOF MEK proteins are unable to recapitulate the effects of *torso^Deg* on transcriptional activity in PBs and PGCs, it was unclear whether this negative result means that a non-canonical Torso-dependent signaling pathway is responsible for activating transcription in *gcl* PBs and PGCs. To explore this question further, we tested whether ectopic expression of GOF MEK can induce defects in the sequestration of pole plasm components. As shown in *Figure 16*, MEK^E203K or MEK^F53S protein induces the inappropriate dispersal of *gcl* and *pgc* mRNAs into the surrounding soma in a pattern very similar to that observed in *torso^Deg* and *gcl* embryos. Thus, this *gcl* phenotype would appear to depend upon the canonical terminal signal transduction cascade.

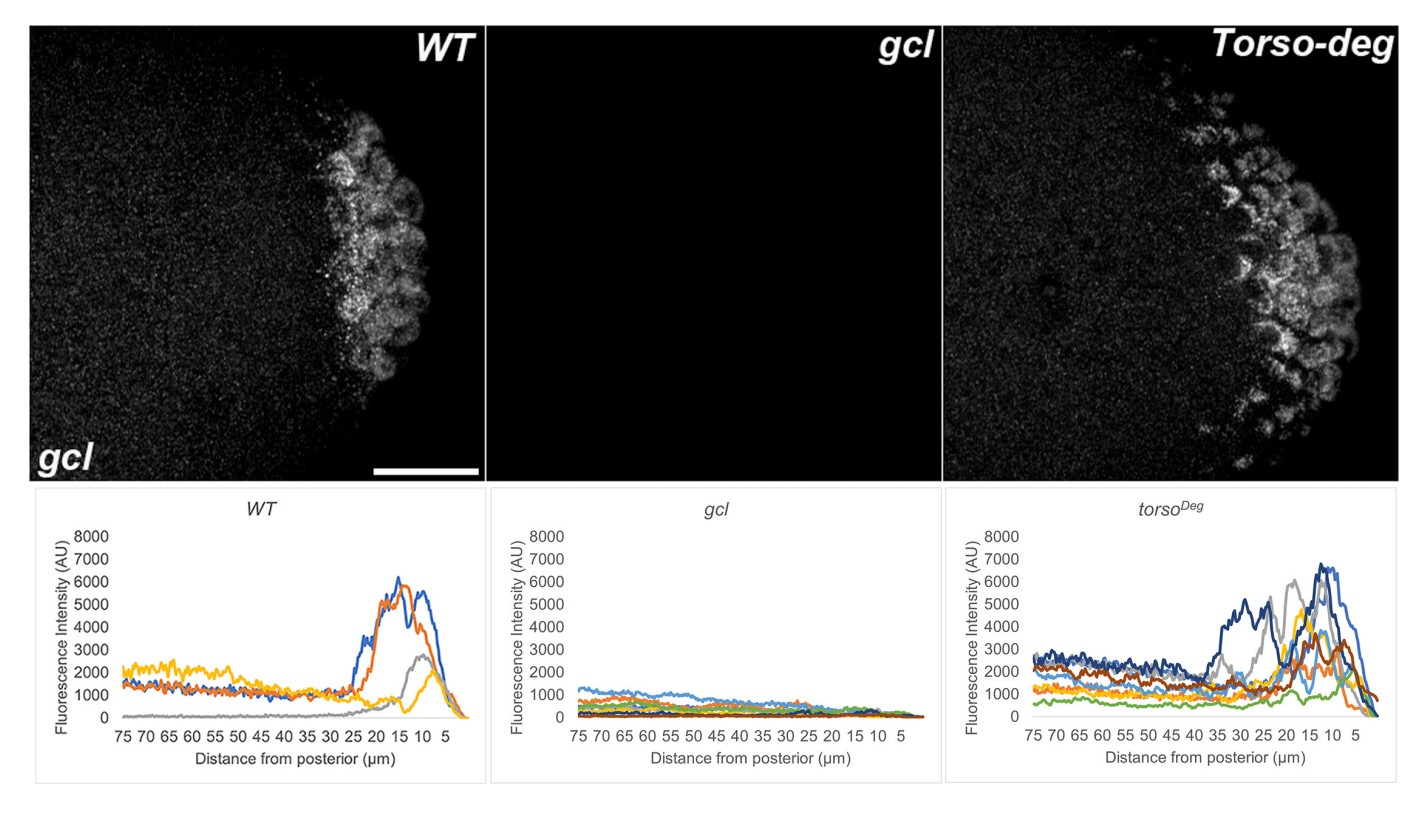

**Figure 12.** *gcl* RNA is mislocalized from the posterior in *torso*[Deg] embryos. smFISH using probes specific for *gcl* was performed in 0–3 hr old embryos to assess whether pole plasm is properly localized in *gcl* and *torso*[Deg] embryos. *gcl* embryos lack *gcl* RNA, as previously reported (*Jongens et al., 1992*). On top, images are representative maximum intensity projections of the posterior pole of each indicated genotype. Scale bar represents 10 µm. Below, plot profiles show mislocalization of pole plasm (visualized using *gcl*) away from posterior cap in *torso*[Deg] embryos (see Materials and methods for details of quantification). Each plot shows a representative experiment, with each line depicting pole plasm distribution of an individual embryo. n = 11, 10, and 16 for wild type (WT), *gcl*, and *torso*[Deg], respectively.

## Discussion

*gcl* differs from other known maternally deposited germline determinants in that it is required for the formation of PBs and PGCs. *gcl* PGCs exhibit a variety of defects during the earliest steps in PGC development. Unlike WT, *gcl* PGCs fail to properly establish transcriptional quiescence. While other genes like *nos* and *pgc* are required to keep transcription shut down in PGCs, their functions only come into play after PGC cellularization (*Deshpande, 2004*; *Deshpande et al., 1999*; *Martinho et al., 2004*). By contrast, *gcl* acts at an earlier stage beginning shortly after nuclei first migrate into the posterior pole plasm and initiate PB formation. In *gcl* PBs, ongoing transcription of genes that are active beginning around nuclear cycle 5–6 is not properly turned off. This is not the only defect in germline formation and specification. As in WT, the incoming nuclei (and the centrosomes associated with the nuclei) trigger the release of the pole plasm from the posterior cortex. However, instead of sequestering the germline determinants in PBs so that they are incorporated into PGCs during cellularization, the determinants disperse into the soma where they become associated with the cytoplasmic territories of nearby somatic nuclei. There are also defects in bud formation and cellularization. Like the release and sequestration of germline determinants, these defects have been linked to the actin cytoskeleton and centrosomes (*Cinalli and Lehmann, 2013*; *Lerit et al., 2017*).

Two models have been proposed to account for the PGC defects in *gcl* mutants. In the first, *Leatherman et al., 2002* attributed the disruptions in PGC development to a failure to turn off ongoing transcription. The second argues that the role of *gcl* in imposing transcriptional quiescence is irrelevant (*Cinalli and Lehmann, 2013*; *Pae et al., 2017*). Instead, the defects are proposed to

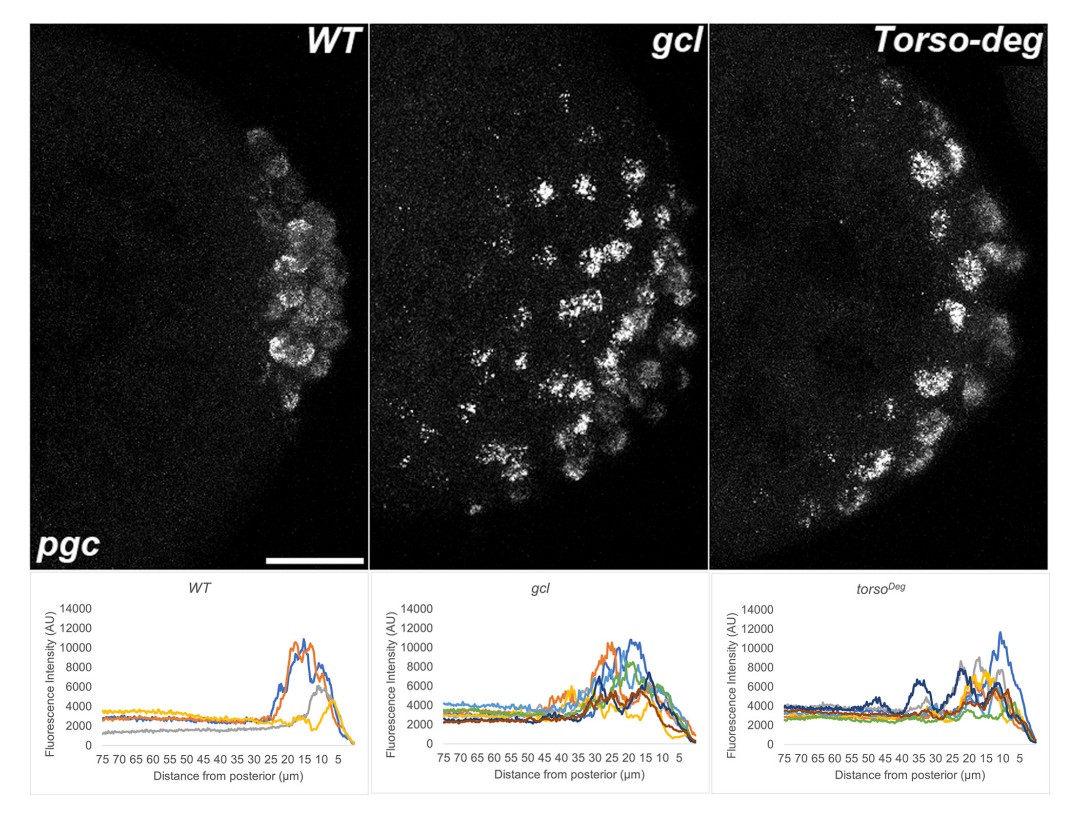

**Figure 13.** *pgc* RNA is mislocalized from the posterior in *gcl* and *torso*^Deg embryos. smFISH using probes specific for *pgc* was performed in 0–3 hr old embryos to assess whether pole plasm is properly localized in *gcl* and *torso*^Deg embryos. On top, images are representative maximum intensity projections of the posterior pole of each indicated genotype. Scale bar represents 10 μm. Below, plot profiles show mislocalization of pole plasm (visualized using *pgc*) away from posterior cap in *gcl* and *torso*^Deg embryos (see Materials and methods for details of quantification). Each plot shows a representative experiment, with each line depicting pole plasm distribution of an individual embryo. n = 10, 14, and 14 for wild type (WT), *gcl*, and *torso*^Deg, respectively.

arise from a failure to degrade the Torso receptor. In the absence of Gcl-dependent proteolysis, high local concentrations of the Tsl ligand modifier at the posterior pole would activate the Torso receptor. According to this model, the ligand–receptor interaction would then trigger a novel, transcription-independent signal transduction pathway in PBs and PGCs that disrupts their development. These conflicting models raise several questions. Does *gcl* actually have a role in establishing transcriptional quiescence in PBs and PGCs? If so, is this activity relevant for PB and PGC formation? Is the stabilization of Torso in *gcl* mutants responsible for the failure to shut down transcription in PBs and PGCs? If not, does *gcl* target a novel, transcription-independent but Torso-dependent signaling pathway? Is the stabilization of Torso responsible for some of the other phenotypes that are observed in *gcl* mutants? In the studies reported here we have addressed these outstanding questions, leading to a resolved model of Gcl activity and function.

We show that shutting off transcription is, in fact, a critical function of Gcl protein. As previously documented by Leatherman et al., we find that several of the key X-linked transcriptional activators of *Sxl-Pe* are not repressed in newly formed PBs and early PGC nuclei, and *Sxl-Pe* transcription is inappropriately activated in the presumptive germline. In previous studies, we found that ectopic expression of *Sxl* in *nos* mutants disrupts PGC specification. In this case, the specification defects in *nos* embryos can be partially rescued by eliminating Sxl activity (*Deshpande et al., 1999*). The same is true for *gcl* mutants: elimination or reduction in Sxl function ameliorates the *gcl* defects in PGC formation/specification. Conversely ectopic expression of Sxl early in embryogenesis mimics the effects of *gcl* loss on PGC formation. Importantly, the role of Gcl in inhibiting *Sxl-Pe* transcription is not dependent upon other constituents of the pole plasm. When Gcl is ectopically expressed at the

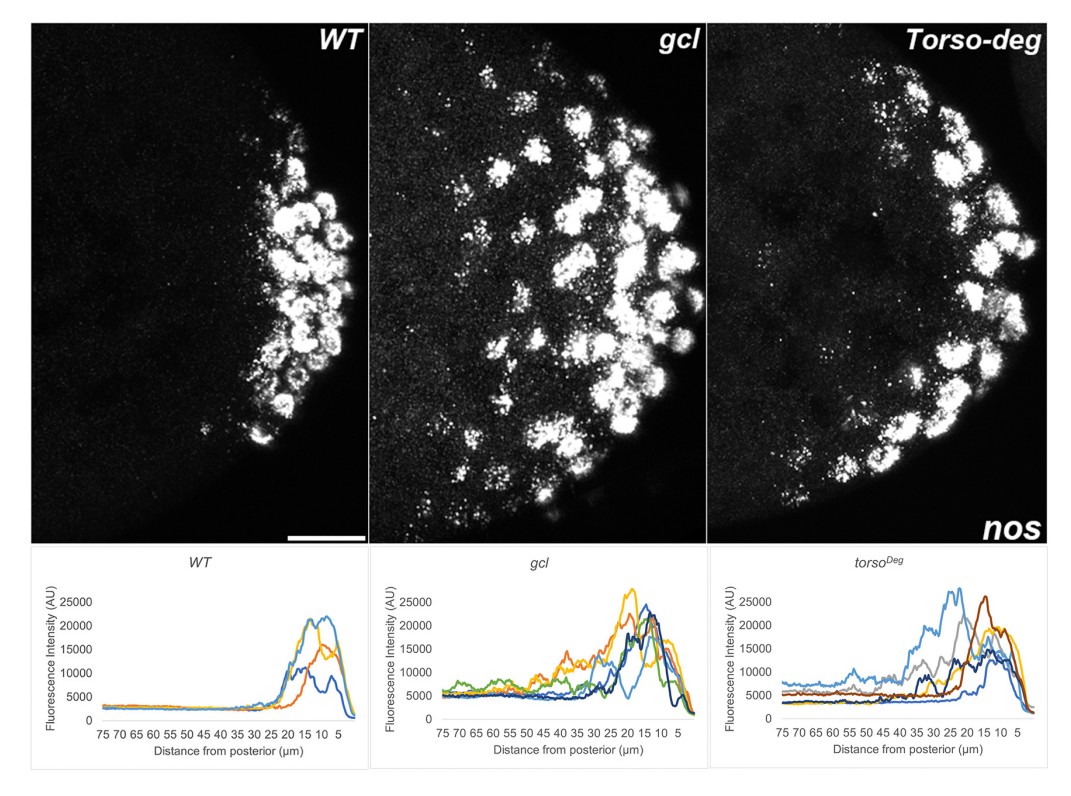

**Figure 14.** *nos* RNA is mislocalized from the posterior in *gcl* and *torso*<sup>Deg</sup> embryos. smFISH using probes specific for *nos* was performed in 0–3 hr old embryos to assess whether pole plasm is properly localized in *gcl* and *torso*<sup>Deg</sup> embryos. On top, images are representative maximum intensity projections of the posterior pole of each indicated genotype. Scale bar represents 10 µm. Below plot profiles show mislocalization of pole plasm (visualized using *nos*) away from posterior cap in *gcl* and *torso*<sup>Deg</sup> embryos (see Materials and methods for details of quantification). Each plot shows a representative experiment, with each line depicting pole plasm distribution of an individual embryo. n = 4, 6, and 6 for wild type (WT), *gcl*, and *torso*<sup>Deg</sup>, respectively.

anterior of the embryo, it can repress *Sxl*. This observation is consistent with the effects of ectopic Gcl on the transcription of other genes reported by *Leatherman et al., 2002*. Since the rescue of *gcl* by eliminating the *Sxl* gene or reducing its activity is not complete, one would expect that there must be other important *gcl* targets. These targets could correspond to one or more of the other genes that are misexpressed in *gcl* PB/PGCs. Consistent with this possibility, transcriptional silencing in *gcl* PBs/PGCs is reestablished when terminal signaling is disrupted by mutations in the *tsl* gene. On the other hand, it is possible that excessive activity of the terminal signaling pathway also adversely impacts some non-transcriptional targets that are important for PB/PGC formation and that transcriptional silencing in only part of the story (see below).

*Pae et al., 2017* showed that mutations in the Gcl interaction domain of Torso (*torso*<sup>Deg</sup>) stabilize the receptor and disrupt PGC formation. Consistent with the notion that Torso receptor is the primary, if not the only, direct target of *gcl*, they found that mutations in the Torso ligand modifier, *tsl*, or RNAi knockdown of *torso* rescued the PGC formation defects in *gcl* embryos. As would be predicted from their findings and ours, ectopic expression of the Torso<sup>Deg</sup> protein induces the inappropriate transcription of *sis-b* and *Sxl-Pe* in PBs and newly formed PGCs. Thus, the failure to shut down ongoing transcription in *gcl* PBs and PGCs must be due (at least in part) to the persistence of the Torso receptor in the absence of Gcl-mediated degradation. Corroborating this idea, the ectopic activation of transcription in *gcl* PGCs is no longer observed when the terminal signaling pathway is disrupted by the removal of *tsl*. Taken together, these data strongly suggest that the establishment/maintenance of transcriptional silencing in PBs is a critical function of Gcl.

Since RNAi knockdowns of terminal pathway kinases downstream of *torso* did not rescue *gcl* mutants, *Pae et al., 2017* postulated that the Tsl-Torso receptor interaction triggered a novel, non-

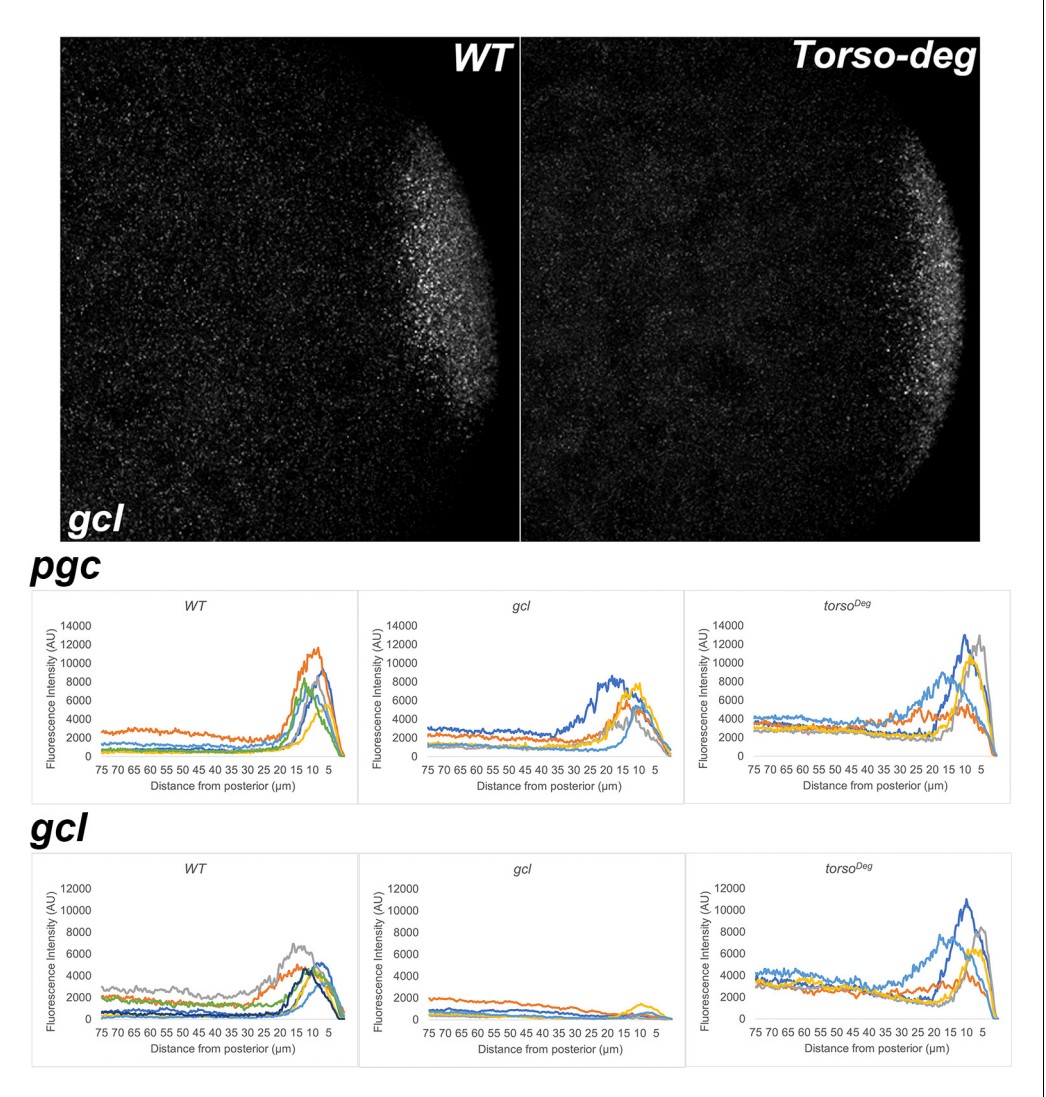

**Figure 15.** Before pole buds develop, pole plasm distribution is unaltered in *gcl* and *torso*[Deg] embryos. smFISH using probes specific for *pgc* or *gcl* was performed in 0–3 hr old embryos to assess whether pole plasm is properly localized in young *gcl* and *torso*[Deg] embryos. Images are representative maximum intensity projections of the posterior pole of each indicated genotype. Scale bar represents 10 μm. Below, plot profiles show proper anchoring and localization of pole plasm (visualized using *pgc* or *gcl*) at the posterior cap in *gcl* and *torso*[Deg] embryos (see Materials and methods for details of quantification). Each plot shows a representative experiment, with each line depicting pole plasm distribution of an individual embryo. For the *pgc* smFISH experiment, n = 9, 7, and 7 for wild type (WT), *gcl*, and *torso*[Deg], respectively. For the *gcl* smFISH experiment, n = 14, 9, and 8 for WT, *gcl*, and *torso*[Deg], respectively.

canonical signal transduction pathway that disrupted PGC development. If their suggestion is correct, then the activation of *sis-b* and *Sxl-Pe* in PBs/PGCs in *gcl* and *torso*[Deg] embryos would be mediated by this novel terminal signaling pathway. Here, our results are ambiguous. Consistent with the suggestion of *Pae et al., 2017*, GOF mutations in MEK, a downstream kinase in the Torso signaling pathway, did not activate *Sxl-Pe* transcription in pole cells. However, an important caveat is that the GOF activity of MEK variants we tested is likely not equivalent to the activity from the normal Torso-dependent signaling cascade (*Goyal et al., 2017*). As the pole plasm contains at least two other factors that help impose transcriptional quiescence, the two GOF MEK mutants we tested may simply not be sufficient to overcome their repressive functions. Two observations are consistent with this possibility. First, like *torso*[Deg], we found that MEK[E203K] induces *Sxl-Pe* expression in male somatic

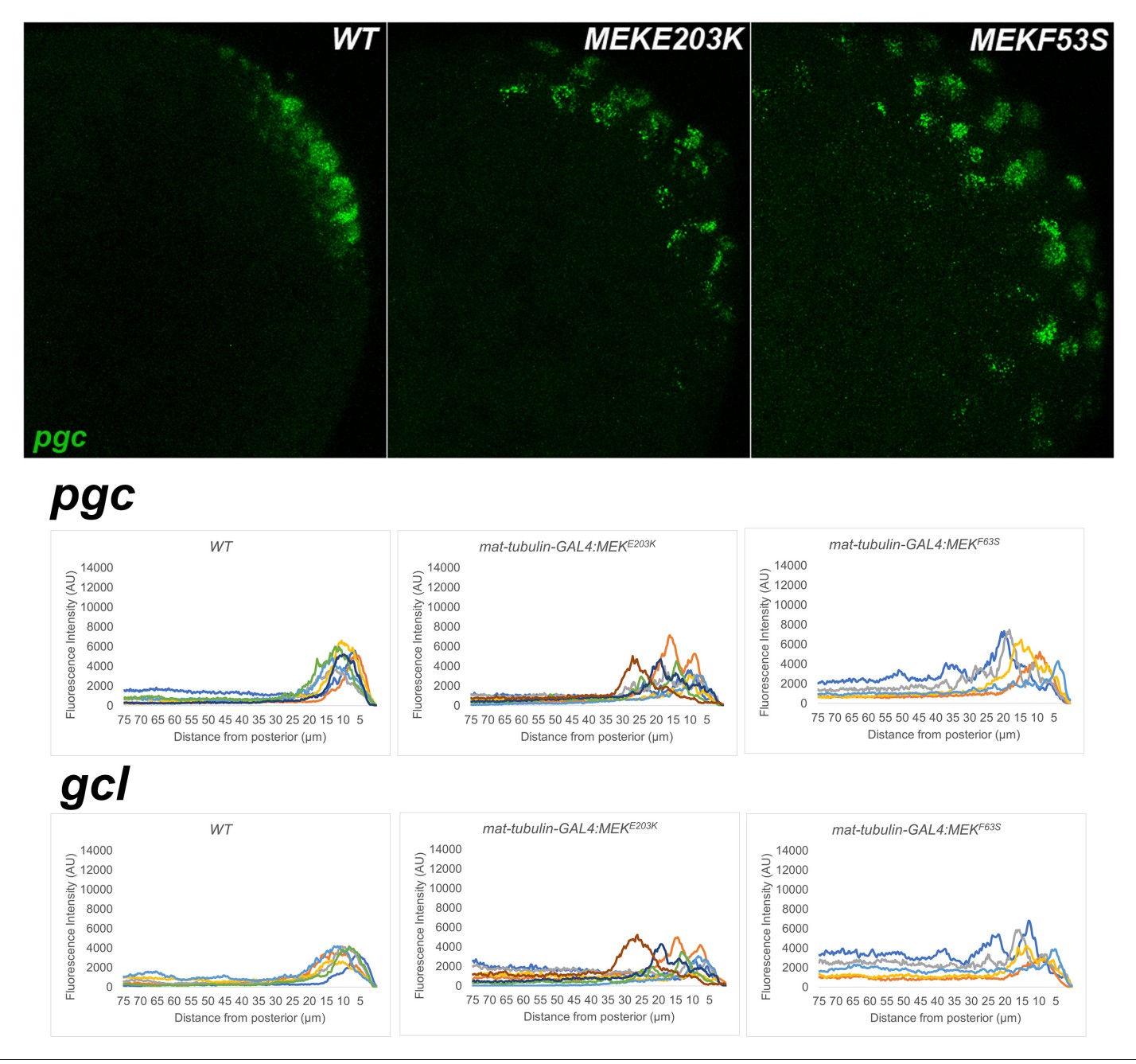

**Figure 16.** MEK gain of function embryos also display defects in pole plasm localization. smFISH using probes specific for *pgc* or *gcl* was performed in 0–3 hr old embryos to assess whether pole plasm is properly localized in embryos collected from mothers expressing MEK[E203K] or MEK[F53S] driven by *mat-GAL4*. On top, images are representative maximum intensity projections of *pgc* RNA localization at the posterior pole of each indicated genotype. Scale bar represents 10 μm. Below, plot profiles show mislocalization of pole plasm (visualized using either *pgc* or *gcl*) away from posterior cap in embryos expressing one of two MEK gain of function transgenes (E203K and F53S) (see Materials and methods for details of quantification). Each plot shows a representative experiment, with each line depicting pole plasm distribution of an individual embryo. n = 15, 19, and 9 for wild type (WT), MEK[E203K], and MEK[F53S], respectively.

nuclei. The same is true for a viable GOF mutation in Torso: it can induce ectopic activation of *Sxl-Pe* in male somatic nuclei, but is unable to activate *Sxl-Pe* in PGCs (*Deshpande, 2004*). Second, a key terminal pathway transcription target *tailless* is not expressed in *gcl* mutant PBs/PGSs even though the terminal pathway should be fully active. This is also true for embryos expressing *torso*[Deg] and the two GOF MEK proteins. For these reasons, we cannot unambiguously determine if it is the

canonical terminal signaling pathway or another, noncanonical signaling pathway downstream of Torso that is responsible for the expression of *sis-b*, *Sxl-Pe*, and other genes in *gcl* mutant PB/PGCs.

There are also reasons to think that the canonical Torso signal transduction cascade must be inhibited for proper PGC formation. One of the more striking phenotypes in *gcl* mutants is the dispersal of key germline mRNA and protein determinants into the surrounding soma. A similar disruption in the sequestration of pole plasm components is observed not only in *torso^{Deg}* embryos but also in *MEK^{E203K}* and *MEK^{F53S}* embryos. Thus, this *gcl* phenotype would appear to arise from the deployment of the canonical Torso receptor signal transduction cascade, at least up to the MEK kinase. However, this result does not exclude the possibility that the Tsl→Torso→ERK pathway has other non-transcriptional targets that, like *Sxl-Pe* expression, can also interfere with PB/PGC formation. If this was the case, it could potentially explain why global transcriptional inhibition failed to rescue the PGC defects in *gcl* embryos (*Cinalli and Lehmann, 2013*). In this respect, a potential—if not likely—target is the microtubule cytoskeleton. In previous studies, we found that the PB and PGC formation defects as well as the failure to properly sequester critical germline determinants in *gcl* arise from abnormalities in microtubule/centrosome organization (*Lerit et al., 2017*). Preliminary imaging experiments indicate that centrosome distribution of *torso^{Deg}* PBs is also abnormal, suggesting that inappropriate activation of the terminal signaling pathway perturbs the organization or functioning of the microtubule cytoskeleton and/or centrosomes. Such a mechanism would also be consistent with the dispersal of germline mRNA and protein determinants in *torso^{Deg}* and GOF *MEK* embryos. While further experiments will be required to demonstrate microtubule and centrosomal aberrations in *torso^{Deg}* and GOF *MEK* embryos, a role for a receptor-dependent MEK/ERK signaling cascade in promoting centrosome accumulation of γ-tubulin and microtubule nucleation has been documented in mammalian tissue culture cells (*Colello et al., 2012*). It is thus conceivable that MEK/ERK signaling has a similar role in Drosophila PB nuclei and PGCs. It will be important to determine if Torso-dependent activation of MEK/ERK can perturb the behavior or organization of centrosomes and/or microtubules in early embryos, and, if so, whether the influence can alter the pole plasm RNA anchoring and/or transmission. Taken together, our data reveal a mutual antagonism between the determinants that specify germline versus somatic identity. Future studies will focus on how and when during early embryogenesis such feedback mechanisms are activated and calibrated to establish and/or maintain germline/soma distinction.

## Materials and methods

### Key resources table

| Reagent type (species) or resource | Designation | Source or reference | Identifiers | Additional information |
|---|---|---|---|---|
| Genetic reagent (D. melanogaster) | *gcl* | *Jongens et al., 1994* | | |
| Genetic reagent (D. melanogaster) | *gcl-bcd-3'UTR* | *Jongens et al., 1994* | | |
| Genetic reagent (D. melanogaster) | *Maternal-tubulin-GAL4 (67.15)* | Eric Wieschaus | | |
| Genetic reagent (D. melanogaster) | *nosGAL4-VP16* | Bloomington Drosophila Stock Center | BDSC: 7303; RRID:BDSC_7303 | |
| Genetic reagent (D. melanogaster) | *UASp-Sxl (DB106)* | Helen Salz | | Maintained in the lab of H. Salz |
| Genetic reagent (D. melanogaster) | *Sxl^{7BO}* | Tom Cline | | |
| Genetic reagent (D. melanogaster) | *UAS-Sxl RNAi (VALIUM20)* | Bloomington Drosophila Stock Center | BDSC: 34393; RRID:BDSC_34393 | |

*Continued on next page*

*Continued*

| Reagent type (species) or resource | Designation | Source or reference | Identifiers | Additional information |
|---|---|---|---|---|
| Genetic reagent (*D. melanogaster*) | *tsl⁴* | Bloomington Drosophila Stock Center | BDSC: 3289; RRID:BDSC_3289 | |
| Genetic reagent (*D. melanogaster*) | *UASp-torsoᴰᵉᵍ* | *Pae et al., 2017* | | Maintained in the lab of R. Lehmann |
| Genetic reagent (*D. melanogaster*) | *MEKᴱ²⁰³ᴷ* | *Goyal et al., 2017* | | Maintained in the lab of S. Shvartsman |
| Genetic reagent (*D. melanogaster*) | *MEKᶠ⁵³ˢ* | *Goyal et al., 2017* | | Maintained in the lab of S. Shvartsman |
| Genetic reagent (*D. melanogaster*) | *UAS-egfp RNAi* (VALIUM20) | Bloomington Drosophila Stock Center | BDSC: 41552; RRID:BDSC_41552 | |
| Antibody | Anti-Vasa (rat polyclonal) | Paul Lasko | RRID:AB_2568498 | Used 1:1000 |
| Antibody | Anti-Vasa (mouse monoclonal) | Developmental Studies Hybridoma Bank | DSHB: 46F11; RRID:AB_10571464 | Used 1:15 |
| Antibody | Anti-Sxl (mouse monoclonal) | Developmental Studies Hybridoma Bank | DSHB: M18; RRID:AB_528464 | Used 1:10 |
| Sequence-based reagent | *pgc* | *Eagle et al., 2018* | smFISH probe set | Exonic probes |
| Sequence-based reagent | *gcl* | *Eagle et al., 2018* | smFISH probe set | Exonic probes |
| Sequence-based reagent | *nos* | *Eagle et al., 2018* | smFISH probe set | Exonic probes |
| Sequence-based reagent | *Sxl* | Thomas Gregor | smFISH probe set | Intronic probes |
| Sequence-based reagent | *sis-b* | Thomas Gregor | smFISH probe set | Intronic probes |
| Sequence-based reagent | *run* | Thomas Gregor | smFISH probe set | Intronic probes |
| Sequence-based reagent | *tll* | Biosearch Technologies; this paper | smFISH probe set | Exonic probes; sequences available in *Supplementary file 1* |
| Other | Hoescht | Invitrogen | Fisher Scientific: H3570 | |

## Fly stocks and genetics

The following fly stocks were used for the analysis reported in this manuscript. *white¹* (*w¹*) was used as the WT stock. *gcl*, a null allele, and *gcl-bcd-3'UTR* stocks were generous gifts from *Jongens et al., 1994*; *Jongens et al., 1992*. *tsl⁴* (BDSC #3289), a loss-of-function mutation, was obtained from Eric Wieschaus. *egfp RNAi* (BDSC #41552), *UAS-Sxl* (Helen Salz - DB106), and MEK gain-of-function transgenic stocks *MEKᴱ²⁰³ᴷ* and *MEKᶠ⁵³ˢ* (gift of Stas Shvartsman, *Goyal et al., 2017*) were driven by *maternal-tubulin-GAL4* (*67.15*) driver stock, which carries four copies of *maternal-tubulin-GAL4* (gift from Eric Wieschaus). The *nosGAL4-VP16* driver (BDSC #7303) was also used. *UAS-torsoᴰᵉᵍ* flies were kindly provided by Ruth Lehmann (*Pae et al., 2017*). The *Sxl* deficiency line, *Sxl⁷ᴮᴼ*, was a gift from Tom Cline.

## Immunostaining

Embryos were formaldehyde-fixed, and a standard immunohistochemical protocol was used for DAB staining as described previously (*Deshpande et al., 1999*). Fluorescent immunostaining employed fluorescently labeled (Alexa) secondary antibodies. The primary antibodies used were mouse anti-

Vasa (1:10, DSHB, Iowa City, IA), rat anti-Vasa (1:1000, gift of Paul Lasko), mouse anti-Sex lethal (1:10, DSHB M18, Iowa City, IA), and rabbit anti-Centrosomin (1:500, gift from Thomas Kaufmann). Secondary antibodies used were Alexa Fluor goat anti-rat 488 or 546 (1:500, ThermoFisher Scientific, Waltham, MA) and Alexa Fluor goat anti-rabbit 647 (1:500, ThermoFisher Scientific, Waltham, MA), DAPI (10 ng/mL, ThermoFisher Scientific, Waltham, MA), and Hoescht (3 µg/mL, Invitrogen, Carlsbad, CA). Stained embryos were mounted using Aqua Poly/mount (Polysciences, Warrington, PA) on slides. At least three independent biological replicates were used for each experiment.

Single molecule fluorescent in situ hybridization smFISH was performed as described by Little and Gregor using formaldehyde-fixed embryos (*Little et al., 2015*; *Little and Gregor, 2018*). All probe sets were designed using the Stellaris probe designer (20-nucleotide oligonucleotides with 2-nucleotide spacing). *pgc*, *gcl*, and *nanos* smFISH probes (coupled with either atto565 or atto647 dye, Sigma, St. Louis, MO) were a gift from Liz Gavis (*Eagle et al., 2018*), and *Sxl*, *sis-b,* and *runt* intronic probes (coupled with either atto565 or atto633 dye, Sigma, St. Louis, MO) were a gift from Thomas Gregor. *tll* probes (coupled with Quasar 570) were produced by Biosearch Technologies (Middlesex, UK). All samples were mounted using Aqua Poly/mount (Polysciences, Warrington, PA) on slides. At least three independent biological replicates were used for each experiment.

## Statistical analysis

For smFISH experiments, total number of embryos expressing *sis-b*, *runt*, or *Sxl* in PBs/PGCs were counted, and pairwise comparisons of the proportion of embryos positive for transcription in PBs/PGCs or proportion of male embryos expressing *Sxl* in the soma were performed using Fisher's exact test. Sex bias in *gcl* and *gcl;Sxl^{RNAi}* embryos was analyzed by comparing proportions also using Fisher's exact test. To calculate significant differences in number of embryos displaying Sxl expression in pole cells or reduced at the anterior from ectopic *gcl* expression (based on DAB-visualization), we used Welch's two sample t-test. Using NC13/14 embryos, PGCs were counted from the first Vasa-positive cell to the last through an entire z-volume captured at 1-micron intervals. Rescue in *gcl;tsl* embryos was analyzed either using Fisher's exact test for proportions of embryos showing PGC transcription or a one-way ANOVA with pairwise t-test comparisons for pole cell counts. Data were plotted and statistical analyses were performed using Microsoft Excel, R Project, or GraphPad Prism software. For the *Sxl* RNAi rescue experiment, data were analyzed by Student's two-tailed t-test or a nonparametric Mann–Whitney U-test and are displayed as mean ± SD. Data shown are representative results from at least two independent biological replicates.

## Microscopy and image analysis

A Nikon-Microphot-SA microscope was used to capture images of DAB-stained embryos (40×). Images for the *Sxl* RNAi rescue experiment were acquired using a 100×, 1.49 NA Apo TIRF oil immersion objective on a Nikon Ti-E system fitted with a Yokagawa CSU-X1 spinning disk head, Hamamatsu Orca Flash 4.0 v2 digital CMOS camera, and Nikon LU-N4 solid state laser launch. Imaging for all other smFISH and fluorescent immunostaining experiments was performed on a Nikon A1 inverted laser-scanning confocal microscope.

Images were assembled using ImageJ (NIH) and Adobe Photoshop and Illustrator software to crop regions of interest, adjust brightness and contrast, generate maximum-intensity projections, and separate or merge channels. To assess the spreading of the RNAs or protein in different mutant backgrounds compared to the control we generated plot profiles using ImageJ. The posterior-most 75 µm of each embryo was plotted for comparison, and embryos from a single biological replicate are plotted in figures given that variation between fluorescence between replicates obscured the pole plasm distribution trends if embryos from all replicates were plotted together.

## Acknowledgements

This work was supported by grants from National Institute of Health (NICHD:093913) to PS and GD, and (NIGMS: 126975) to PS. Work in the Lerit lab was supported by K22HL126922 and R01GM138544. MC was supported by NSF Graduate Research Fellowship (DGE-1656466). We thank Dr. Gary Laevsky and the Molecular Biology Confocal Microscopy Facility which is a Nikon Center of Excellence. Gordon Grey provided fly media. We thank the Bloomington stock

center for different fly lines. We gratefully acknowledge Liz Gavis and Ruth Lehmann for advice and reagents during the course of this work.

## Additional information

### Funding

| Funder | Grant reference number | Author |
|---|---|---|
| National Institute of General Medical Sciences | 126975 | Paul Schedl |
| Eunice Kennedy Shriver National Institute of Child Health and Human Development | 093913 | Paul Schedl<br>Girish Deshpande |
| National Heart, Lung, and Blood Institute | K22HL126922 | Dorothy A Lerit |
| National Institute of General Medical Sciences | 138544 | Dorothy A Lerit |
| National Science Foundation | DGE-1656466 | Megan M Colonnetta |

The funders had no role in study design, data collection and interpretation, or the decision to submit the work for publication.

### Author contributions

Megan M Colonnetta, Conceptualization, Formal analysis, Funding acquisition, Investigation, Visualization, Writing - original draft, Writing - review and editing; Lauren R Lym, Lillian Wilkins, Formal analysis, Investigation; Gretchen Kappes, Elias A Castro, Pearl V Ryder, Investigation; Paul Schedl, Conceptualization, Supervision, Funding acquisition, Writing - original draft, Writing - review and editing; Dorothy A Lerit, Girish Deshpande, Conceptualization, Supervision, Funding acquisition, Investigation, Visualization, Writing - original draft, Writing - review and editing

### Author ORCIDs

Megan M Colonnetta (iD) https://orcid.org/0000-0001-5685-1670
Lauren R Lym (iD) http://orcid.org/0000-0001-5039-2303
Elias A Castro (iD) http://orcid.org/0000-0002-1439-5918
Dorothy A Lerit (iD) https://orcid.org/0000-0002-3362-8078
Girish Deshpande (iD) https://orcid.org/0000-0002-5200-7090

### Decision letter and Author response

Decision letter https://doi.org/10.7554/eLife.54346.sa1
Author response https://doi.org/10.7554/eLife.54346.sa2

## Additional files

### Supplementary files

• Supplementary file 1. Sequences for smFISH probes complementary to *tailless* exons (designed using Stellaris probe designer).
• Transparent reporting form

### Data availability

All data generated or analyzed during this study are included in the manuscript and supporting files. Source data files have been provided for Figures 3,5,6, and 8.

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
