## [Decision Letter]

**Acceptance summary:**

The paper reconciles conflicting models for the role of the *Drosophila* protein Germ cell-less (Gcl) in the formation of primordial germ cells. The authors provide compelling evidence that the protein establishes/maintains germ-line cells in a quiescent state by interfering with the ability of Torso receptor signaling to activate transcription of a number of genes, the sex determination gene Sex-lethal being the main focus here. This work also demonstrates a role for Gcl and the inhibition of Torso signaling in the proper localization of *Drosophila* germ plasm, suggesting a previously unappreciated role for Torso signaling in the organization or function of the cytoskeleton. This paper will be of specific interest to investigators who study germ and stem cell formation, and more broadly, to those concerned with the question of how receptor-mediated signaling, and more specifically, signaling via receptor tyrosine kinases, influences transcription and other cellular processes.

**Decision letter after peer review:**

[Editors’ note: the authors submitted for reconsideration following the decision after peer review. What follows is the decision letter after the first round of review.]

Thank you for submitting your work entitled "Antagonism between Germ cell-less and Torso regulates transcriptional quiescence underlying germline/soma distinction" for consideration by *eLife*. Your article has been reviewed by a Senior Editor and two reviewers. The reviewers have opted to remain anonymous. Our decision has been reached after consultation between the reviewers.

Based on these discussions and the individual reviews below, we regret to inform you that your work will not be considered further for publication in *eLife*. As you will see the reviewers raise issue, some of which I will call out here:

1) They do not feel that the experiments convincingly establish hypothesis set up about the work of Pae et al.

2) They found the subtlety of the quantitative results to be inconsistent with a simple explanation.

3) After detailed review, they judge that the work will be of more interest to specialists than to a more general readership. It may therefore be more appropriate for Development.

4) In several places they (and I) found the manuscript hard to follow. Suggestions to address this are included in the reviews.

Reviewer #1:

This paper helps resolve heretofore confusing data on the roles of pole plasm components and the terminal (torso) signaling pathway on germ cell specification, transcriptional quiescence, and maintenance. Specifically, the work helps elucidate the role of the pole plasm component germ cell less (*gcl*). The topic is a good contribution to the field, although of most interest to readers interested in *Drosophila* pole plasm and germ cell specification. One consideration is the method of quantification. The authors score the changes with genotype on a per embryo basis, rather than on a per pole bud/cell basis. However, the number of pole cells formed per embryo varies with the different genotypes assessed. In at least some cases the effect of this means that the effects are likely even stronger than the authors state and the authors are being conservative. However, these points, the rational for the quantification methods used, and whether they might in some other cases overestimate the effects of the genetic tests are not adequately discussed.

1) Abstract: It is hard to grasp the proposed regulatory pathway from the string of contradictions and observations listed. This is made worse by ambiguous statements. For example, does "like *gcl*" mean loss of function or forced overexpression? The two suggest opposite signs in terms of regulation. A simple statement of the proposed regulatory pathway suggested by the new data would help the reader and could replace the last sentence, which has little information content.

2) Introduction: Please clarify here at which nuclear cycle nuclei migrate into the pole plasm. In that context, is very early transcriptional activation (nuclear cycles 6-7) also turned on in nuclei destined for the germ line? If so is that early transcription turned off in the nuclei once they reach the pole plasm? Alternatively, is "early" transcription always kept off in nuclei that will come to inhabit the germ cells? IE: is transcription "switched off" as stated, or is it always maintained in an off state in the nuclei that will come to inhabit the germ plasm?

3) Introduction, in several places: Should "*gcl* mutant embryos" instead be "embryos derived from *gcl* mutant mothers"? The authors should take care with the wording and the labeling of figures throughout in this regard.

4) Introduction: The authors say the "…number correlated well…". However, it is actually reciprocal. Embryos from *gcl* homozygous mutant mothers had a high percent of pole bud nuclei with phosphorylated CTD but a low number of pole cells compared to wild type.

5) Subsection “Gcl represses the expression of XCEs in nascent PGCs”: Please define XCE at first usage for those not in the immediate field. I suggest that the authors move the first paragraph of the Results into the Introduction. Since Sxl-PE is only expressed in early female embryos, do the effects of having a *gcl* mutant mother on firing of Sxl-PE only show up in female but not in male embryos?

6) Figure 1 and Subsection “Gcl represses the expression of XCEs in nascent PGCs”: To convince the reader that sisb and Sxl are being newly transcribed the authors should make clear that they are only scoring hybridization signal in dots in the nucleus. Were the data shown and scored from an intronic probe? That would be best. In stating the quantitation of 67% and 42%, the authors say embryos. Was signal only observed in one or two pole cells per embryo, as the figures shown suggest? Should the authors also give numbers for what percent of pole buds and pole cells showed signal (scoring over a number of embryos)? Especially since the actual number of pole buds/pole cells per embryo is different in the different genotypes. This is a point for the quantitation in many of the figures. At least the authors should lay out the rational for counting on a per embryo vs. a per pole cell basis, and discuss how the differences in number of germ cell precursors per embryo in the different genotypes might effect the meaning of the counts.

7) Subsection “*Sxl* RNA is detected in *gcl* pole buds and PGCs”: The authors should give the frequencies (on a per cell/bud basis) for male vs female and indicate if the numbers are statistically significantly different.

8) Subsection “*Sxl* RNA is detected in *gcl* pole buds and PGCs”: Here the numbers given are for PGC nuclei rather than for embryos. That is good. But the authors should state how many different embryos were assessed and wether all all PGCs in each embryo were counted. Were the counts per section or obtained by focusing up andf down on a whole mount and so counting all PGCs is each embryo.

9) Subsection “Ectopic expression of *gcl* represses *Sxl*”: Were only female embryos scored? Why did obly 9/13 embryos show reduction of Sxl at the anterior? How did the authors ascertain on an embryo by embryo basis that reduction of Sxl in the anterior was coincident with expression of *gcl* from the gCl^-^bcd 3"UTR transgene? Did the 4 embryos that did not show reduction of anterior Sxl also not express *gcl* at the anterior? The result would be more convincing if the authors showed co-staining for expression of Gcl protein in the same embryos.

10) Subsection “Premature expression of *Sxl* in the PGCs leads to germ cell loss and defective germ cell migration”: I thought that Lehmann showed that nosGal4-VP16 did not drive transcription in early pole cells? The authors should show data backing up their statement that the number of PGCs was reduced in nosGal4-VP16; UAS-Sxl early embryos. That reduction is not apparent in the later embryos shown in Figure 4. How early does the ectopic Sxl become expressed under the conditions used?

11) Subsection “Simultaneous removal of *gcl* and *Sxl* ameliorates the *gcl* phenotype” and Figure 5: The data would be more convincing if the authors had some independent way to classify male vs. female embryos, other than expressing or not expressing Sxl. Could they use a balancer with GFP or some other way to distinguish male control with intact Sxl from female embryos with Sxl deleted?

12) Subsection “A degradation-resistant form of Torso also activates transcription in PGCs”, and data in Figure 7: Again, please clarify the quantitation. Does 27% of torsodeg embryos express Sisb meant that all PGCs in that positive embryo have Sisb or at least one Pgc per embryo? The two have very different implications as to the strength of the effect. Might it be better to give counts on the percent of PGCs that score positive (counting at least 10 different embryos). What are the corresponding numbers for the wild type controls in this experiment? Panel A (wt) shows that some PGCs express Sisb. How significant is the difference between control and experimental for each?

13) Discussion: Could the effects on transcription be secondary to a failure to confine pole plasm components to the pole cell buds?

Reviewer #2:

Past work suggests that the role of the *Drosophila* Germ Cell-Less (GCL) is to maintain primordial germ cells in a transcriptionally quiescent state. Pae et al., 2017 found that GCL mediates degradation of the Torso RTK, whose ectopic activity perturbs pole cell development. Colonetta et al., confirm that some somatic genes are ectopically activated in *gcl* mutant pole buds/cells and that ectopic GCL can inhibit expression of some genes in somatic cells, as can a mutant version of Torso (from Pae et al.,) that is not degraded by GCL. Contradicting Pae et al., Colonetta et al., argue that Torso's effects on PGCs require MAP Kinase pathway signaling. They also describe a novel phenotype in Torso[deg], *gcl* mutant, and MEK gain-of-function MEK embryos where pole plasm components get mis-localized. My criticisms are as follows: Several of the experiments contradicting/correcting Pae et al., are not definitive. Moreover, even if correct, the findings refine the model of Pae et al., without providing much new mechanistic insight. While of interest to *Drosophila* workers, particularly ones with an interest in early patterning and primordial germ cells, this manuscript might be considered esoteric by others who may also find reading difficult, owing to some poor writing.

1) The authors clearly show a number of genes, particularly ones involved in sex determination (sis-b, runt, Sxl), are expressed ectopically in polar buds/primordial germ cells, in the absence of GCL. While the authors add some additional genes to the list, this phenomenon has previously been demonstrated by these and other authors, and as noted in this manuscript, although the use here of smFISH to detect nascent transcripts is a nice touch.

2) The authors express *gcl* ectopically at the anterior of the embryo, showing that this leads to a local reduction in the expression of Sxl and use this finding to argue that GCL inhibits transcription of specific targets, and is sufficient to do so, "even in the absence of other pole plasm components," presumably including Torso. However, Torso is active in this domain and if Torso does lead to the activate of transcription Sxl via its effects upon sis-a, sis-b, runt, then this might be the expected result. If indeed the effect of Torso on pole cells is transcriptional, then Torso may be the sole target of GCL. The definitive experiment to answer this question has not been carried out by these authors and that is to assess the transcription of sis-a, sis-b, runt, Sxl, in pole buds/cells of embryos that are both null mutants for *gcl* and torso. If all of those genes failed to be expressed in the double mutant background, this would provide support for the notion that Torso acts to mediate the transcription of those genes, and might be the sole target of GCL. If any of those genes were expressed, that would argue that GCL can act to silence at least some genes, independent of Torso. Given the close chromosomal linkage of *gcl* and torso, it would be difficult, though not impossible, to obtain double mutants to test. It is certainly possible that with a concerted effort, these double mutants could be isolated and examined within a couple of months. Although the gcl tor/gcl tor double mutant would be ideal, the *gcl*/*gcl*; *tsl*/*tsl* strain might suffice, although we really do not have a complete picture of Tsl's role in the activation of Torso.

3) With regard to a transcriptional effect of Torso on pole buds/cells, the authors subtly misrepresent Pae et al. to set up a straw man in the statement that Pae et al., "proposed that activated Torso must inhibit PGC via distinct non-canonical mechanism that is both independent of the standard signal transduction and does not involve transcriptional activation." Although Pae et al., do state at several points in their paper that this effect of Torso is not transcriptional in nature, at the end of the manuscript they acknowledge that "additional downstream pathways (to MAPK/ERK) may require ligand-induced Torso receptor activation. Candidates include the JAK/STAT pathway, which is activated in dominant gain-of-function alleles of Torso (Li et al., 2002)." This would clearly be a transcriptional effect. Moreover, there is good reason to suspect that STAT may play a role in Torso's effects on pole buds/cells. First, there are several other contexts in which RTK's activate STAT. More to the point, however, the JAK/STAT pathway has been implicated in the control of Sxl transcription (Sefton et al., 2000), the mRNAs encoding the JAK/STAT pathway components are maternally loaded into the embryo, and Torso-dependent activation of STAT and Ras at an early step in PGC development has been shown to be required for their proliferation and migration (Li et al., 2003). Some of these reports should be of interest to Colonetta et al.

4) The section of the paper on "Simultaneous removal of *gcl* and *Sxl* ameliorates the *gcl* phenotype" is poorly written, probably impenetrable to anyone outside of the community of *Drosophila* workers studying early patterning events. The discussion of the expected progeny and the interpretation of the results are altogether unclear. Similarly, the experiment in which RNAi directed against Sxl is performed in the absence of GCL reveals an unexpected bimodal distribution of almost normal and pole cell-lacking embryos, for which no potential explanation is offered. In the absence of a credible potential explanation of this result, the conclusion that the loss of SXL is suppressing the lack of GCL is not convincing.

5) To ask the question of whether Map Kinase signaling downstream of Torso activity is required for Torso's effect upon pole buds/cells, Pae et al., performed RNAi against dsor1 (MEK) and rolled (MAPK) in a *gcl* null background. The failure of the RNAi treatment to suppress the effect of GCL loss on pole cells, was used as evidence that MAPK signaling downstream of Torso was not required for the Torso-dependent effects on pole buds/cells. To address this question, Colonetta el al., expressed activated MEK in mothers and assessed Sxl expression. Consistent with Pae et al., activated MEK failed to drive Sxl expression. However, they did observe Sxl expression in the somatic cells of some males. Given that this is a contrived and very different situation than pole cells, and that Pae's experiments in pole cells tested for requirement, while Colonetta's tested for sufficiency (apples to oranges) the question of MAPK signaling in Torso's effect upon pole buds/cells remains unanswered and it is baffling why Colonetta et al., did not examine sis-a, sis-b, runt, Sxl expression in the pole buds/cells of *gcl* mutant, dsor1 (MEK) and rolled (MAPK) knockdown embryos.

6) Figure 9, Figure 10, Figure 11, Figure 12, and Figure 13 show plot profiles of various pole plasm components (Vasa, *gcl* RNA, pgc RNA, nos RNA) in individual wild-type, *gcl* mutant and Torso-deg embryos. While the photographic images of the pole plasm components are consistent with the interpretations proposed by the authors, the plot profiles, which apparently only show a subset of the examined embryos for each component/genotype are more effective in displaying embryo to embryo variation, that they are in showing consistent patterns of mis-localization associated with the pole plasm components in the three genetic backgrounds.

[Editors’ note: further revisions were suggested prior to acceptance, as described below.]

Thank you for resubmitting your work entitled "Antagonism between Germ cell-less and Torso regulates transcriptional quiescence underlying germline/soma distinction" for further consideration by *eLife*. Your revised article has been evaluated by Michael Eisen as the Senior and Reviewing Editor.

Following your appeal, the manuscript was re-evaluated. We appreciate that you have addressed the earlier critiques, and made the manuscript significantly clearer. We also note the addition of experiments showing that the transcription of Sxl and sis-b elicited in the absence of Gcl is suppressed by the elimination of Torso activation, providing additional support for their conclusions regarding the connection between Gcl and Torso.

It is our judgment now that the manuscript likely warrants publication in *eLife* if you can address the following remaining concerns raised by one of the reviewers:

1) The distinction between terminal and non-terminal nuclei with respect to the ability of ectopic anterior Gcl to inhibit Sxl transcription should be explained more fully. All of the instances of ectopic Sxl transcription reported here occur in situations in which Torso or MEK signaling are occurring. That being the case it is not possible to conclude unambiguously that Gcl is uniquely able to suppress Sxl transcription. It's ability to suppress Sxl transcription may rely on Sxl transcription having been activated by ectopic Torso or MEK activity. While I would not require the authors to carry out additional experiments in which Gcl is expressed outside of the terminal regions of wild-type female embryos, or even better, in Torso-lacking female embryos, I believe that it is at least essential to discuss this issue and how it affects the conclusion that Gcl is uniquely able to suppress Sxl transcription. On the other hand, the results of examining Sxl transcription in Torso mutant-derived female embryos ectopically expressing Gcl might prove illuminating, if the authors are willing to undertake the effort.

2) The means of expression of the various transgenic construct used should be explained more clearly with respect to whether the constructs are expressed in the female under Gal4-mediated control and deposited in the embryo versus maternal deposition of Gal4 in the embryo and zygotic expression of the gene that has been crossed in. Similarly, the identities of the transgenic constructs, UASt, UASp, or UASother-based, should be noted.

3) The section in the Discussion on the effects of activated MEK, and of the ERK/MAPK cassette on Sxl transcription in PBs/PGCs versus somatic cells should be modified with additional consideration of the possibility that an alternative pathway downstream of Torso does exist, possibly working independent of the MAPK/ERK pathway. In modifying the text, see the more detailed consideration of this possibility in the Public Review, which outlines my reasoning.

---

## [Author Response]

[Editors’ note: The authors appealed the original decision. What follows is the authors’ response to the first round of review.]

Reviewer #1:This paper helps resolve heretofore confusing data on the roles of pole plasm components and the terminal (torso) signaling pathway on germ cell specification, transcriptional quiescence, and maintenance. Specifically, the work helps elucidate the role of the pole plasm component germ cell less (gcl). The topic is a good contribution to the field, although of most interest to readers interested in Drosophila pole plasm and germ cell specification. One consideration is the method of quantification. The authors score the changes with genotype on a per embryo basis, rather than on a per pole bud/cell basis. However, the number of pole cells formed per embryo varies with the different genotypes assessed. In at least some cases the effect of this means that the effects are likely even stronger than the authors state and the authors are being conservative. However, these points, the rational for the quantification methods used, and whether they might in some other cases overestimate the effects of the genetic tests are not adequately discussed.

We realize that the rationale behind the quantitation was not adequately explained in the earlier version. As the reviewer correctly pointed out, we were concerned about the differences between the number of pole buds/cells per embryo among the different genotypes. It is important to note that smFISH technique reports on the stochastic nature of early embryonic transcription and as a result, it is possible that the effects are underestimated. Nonetheless, our observations are statistically highly significant, and the conclusions are robust in all cases. Please see our response to specific point #6 for a more detailed description of our reasoning. We have also included the rationale for this method of quantification in the text in subsection “*Sxl* RNA is detected in *gcl* pole buds and PGCs”.

1) Abstract: It is hard to grasp the proposed regulatory pathway from the string of contradictions and observations listed. This is made worse by ambiguous statements. For example, does "like gcl" mean loss of function or forced overexpression? The two suggest opposite signs in terms of regulation. A simple statement of the proposed regulatory pathway suggested by the new data would help the reader and could replace the last sentence, which has little information content.

We have edited the Abstract suitably to clarify several points. We have specifically used the phrase “embryos maternally compromised for *germ cell-less*” to explain the maternal effect of *gcl* mutation. We have also mentioned this explicitly in the text.

2) Introduction: Please clarify here at which nuclear cycle nuclei migrate into the pole plasm. In that context, is very early transcriptional activation (nuclear cycles 6-7) also turned on in nuclei destined for the germ line? If so is that early transcription turned off in the nuclei once they reach the pole plasm? Alternatively, is "early" transcription always kept off in nuclei that will come to inhabit the germ cells? IE: is transcription "switched off" as stated, or is it always maintained in an off state in the nuclei that will come to inhabit the germ plasm?

There is no reason to believe that specific nuclei are determined to acquire germ cell fate. Exposure and sustained association with the germ plasm components are sufficient to confer pole cell identity. Ephrussi and Lehmann, (1992) demonstrated that ectopic localization of Oskar is sufficient to form functional pole cells in the anterior (Osk-Bcd3’UTR). Importantly, Paul Macdonald and colleagues showed that after heat shock an *hsp-70* promoter-driven *oskar* transgene can form germ cells at random ectopic locations (Ha et al., 1992). These observations demonstrate that, in principle, any somatic nucleus can acquire germ cell fate upon sustained exposure to germ plasm.

Nuclei enter the germ plasm around nuclear cycle 9-10. There is a low level of transcription across the embryo at this stage including the X chromosome counting elements (but not *Sxl-Pe*). Consequently, when they migrate to the germ plasm, nuclei are “transcriptionally” active; however, transcription is then shut down by germ plasm components including *gcl, pgc*, and *nos*. Leatherman and Jongens, (2000) have suggested that Gcl activity is especially crucial in this regard. This is clarified in the text.

3) Introduction, in several places: Should "gcl mutant embryos" instead be "embryos derived from gcl mutant mothers"? The authors should take care with the wording and the labeling of figures throughout in this regard.

We refer to the embryos derived from homozygous *gcl* mothers as “*gcl*”. Pole cells or pole buds of *gcl* embryos are also denoted as such. We have edited the text and the legends carefully to correct this. We explicitly state this early in the Introduction.

4) Introduction: The authors say the "…number correlated well…". However, it is actually reciprocal. Embryos from gcl homozygous mutant mothers had a high percent of pole bud nuclei with phosphorylated CTD but a low number of pole cells compared to wild type.

We thank the reviewer for pointing this out and have corrected the text as recommended.

5) Subsection “Gcl represses the expression of XCEs in nascent PGCs”: Please define XCE at first usage for those not in the immediate field. I suggest that the authors move the first paragraph of the Results into the Introduction. Since Sxl-PE is only expressed in early female embryos, do the effects of having a gcl mutant mother on firing of Sxl-PE only show up in female but not in male embryos?

We have tried to provide adequate background about somatic sex determination pathway to make the text readily accessible to the readers.

Sex-specificity: *gcl* is a maternally deposited RNA, which is translated in a sex-non-specific manner. The mutant phenotype induced by maternal loss of *gcl* i.e. loss of pole buds/cells displays no obvious sex-specificity. It should be noted that in wild type embryos, *Sxl-Pe* transcription is not turned on either in male or female PGCs. *Sxl-Pe* is transcriptionally activated only in female somatic nuclei while it remains off in male somatic nuclei. Our results show that both male and female PGCs showed ectopic activation of *Sxl-Pe* transcription with a modest female specific bias. We have provided quantitation to document this observation. This is consistent with the fact that several of the known Gcl targets are X-linked, including *sis-a*, *sis-b*, *runt* and *Sxl*. The difference in gene dose plus the fact that *sis-a*, *sis-b* and *runt* are X-linked “numerators” likely results in higher (or more frequent) *Sxl-Pe* transcription in female embryos.

6) Figure 1 and Subsection “Gcl represses the expression of XCEs in nascent PGCs”: To convince the reader that sisb and Sxl are being newly transcribed the authors should make clear that they are only scoring hybridization signal in dots in the nucleus. Were the data shown and scored from an intronic probe? That would be best. In stating the quantitation of 67% and 42%, the authors say embryos. Was signal only observed in one or two pole cells per embryo, as the figures shown suggest? Should the authors also give numbers for what percent of pole buds and pole cells showed signal (scoring over a number of embryos)? Especially since the actual number of pole buds/pole cells per embryo is different in the different genotypes. This is a point for the quantitation in many of the figures. At least the authors should lay out the rational for counting on a per embryo vs. a per pole cell basis, and discuss how the differences in number of germ cell precursors per embryo in the different genotypes might effect the meaning of the counts.

This is a good point, and we have edited the text to clarify that we are only scoring embryos based on hybridization signal in PB/PGC nuclei. We have also explained our rationale for scoring embryos rather than PBs/PGCs of each embryo, which we settled on largely due to the stochastic nature of detection of transcription during early embryogenesis.

Somatic transcription during cycle 14, the major wave of ZGA, is characterized by bursting. Even at this stage where transcription is maximally induced, one can observe nuclei in which one or the other copy of a gene, which should be active, is not transcriptional engaged. Earlier, during nuclear cycles 9-11, during PGC formation, levels of transcription are significantly lower than during the major wave of ZGA. At these earlier stages, intervals between transcriptional bursts are expected to much greater, and in static images of ongoing transcription, one expects to see many fewer nuclei in which transcriptionally active genes can be detected. This is the case in WT somatic nuclei during this period, and it is also the case in *gcl* PBs and nascent PGCs: only a subset of the nuclei is actively engaged in transcription. The effects of bursting are reflected in smFISH experiments in which we probed for two different transcripts. In some cases, both genes are found to be active in the same nuclei, while in other nuclei, only one (or no) gene is active. In the case of *sis-b*, the transcription unit is only 1.4 kb. This means that there will be only a short delay between the end of the burst, and the disappearance of the transcript. For *Sxl-Pe* we have intron probes. Introns are spliced out co-transcriptionally, and the signal will disappear even though the gene might still be transcribed. Finally, for PBs, *gcl* embryos have only 1-3 buds as opposed to 5-6 in WT. This means that we can only score a few cells in each embryo. For these reasons, we scored the percent of embryos (WT, *gcl*, Torso^deg^) in which ongoing transcription is detected. This is a fully robust test as *sis-b* and *Sxl-Pe* transcripts are *not* detected in WT PB or PGCs.

We have included a summary of this rationale in the appropriate section where smFISH experiments are first described (subsection “*Sxl* RNA is detected in *gcl* pole buds and PGCs”).

7) Subsection “Sxl RNA is detected in gcl pole buds and PGCs”: The authors should give the frequencies (on a per cell/bud basis) for male vs female and indicate if the numbers are statistically significantly different.

We have repeated this analysis and analyzed the data to assess the sex of the embryos by counting the number of dots of hybridization per nucleus. We were able to sex embryos using number of dots of *Sxl*-specific hybridization (Avila and Erickson, 2007). We also employed smFISH using *sis-B* probe as an additional marker of sexual identity for confirmation. A clear visualization of probe puncta in female vs. male somatic nuclei is presented in Figure 7 (newly added).

8) Subsection “Sxl RNA is detected in gcl pole buds and PGCs”: Here the numbers given are for PGC nuclei rather than for embryos. That is good. But the authors should state how many different embryos were assessed and wether all all PGCs in each embryo were counted. Were the counts per section or obtained by focusing up andf down on a whole mount and so counting all PGCs is each embryo.

All the pole cell nuclei per embryo were counted (n~15); we have included this information for each quantification.

9) Subsection “Ectopic expression of gcl represses Sxl”: Were only female embryos scored? Why did obly 9/13 embryos show reduction of Sxl at the anterior? How did the authors ascertain on an embryo by embryo basis that reduction of Sxl in the anterior was coincident with expression of gcl from the gCl^-^bcd 3"UTR transgene? Did the 4 embryos that did not show reduction of anterior Sxl also not express gcl at the anterior? The result would be more convincing if the authors showed co-staining for expression of Gcl protein in the same embryos.

As males do not express Sxl protein, Sxl staining is observed only in female embryos, so only female embryos have been scored, a point we now clarify in the figure legend/Methods. It should be noted that all female embryos expressing *gCl^-^bcd-3’UTR* showed some reduction in Sxl-specific staining. Also, all male embryos are devoid of Sxl and thus provided an internal negative control. We attempted to use the Gcl antiserum that was kindly provided by the Jongens lab. Unfortunately, there was considerable background staining and those data were omitted.

10) Subsection “Premature expression of Sxl in the PGCs leads to germ cell loss and defective germ cell migration”: I thought that Lehmann showed that nosGal4-VP16 did not drive transcription in early pole cells? The authors should show data backing up their statement that the number of PGCs was reduced in nosGal4-VP16; UAS-Sxl early embryos. That reduction is not apparent in the later embryos shown in Figure 4. How early does the ectopic Sxl become expressed under the conditions used?

We apologize if this was unclearly presented, and we have revised the corresponding legend to appropriately emphasize these relevant results. The *nos-Gal4-VP16* driver was used to drive Sxl misexpression in the mid-to-late stages of embryogenesis. For the early embryos we used a *maternal-tubulin Gal4* driver strain i.e. *67.15* that carries 4 copies of the insert and can drive robust expression. In our hands, substantial deposition of GAL4 protein is more effective than *nos-Gal4-VP16* to partially overcome the transcriptional silencing in PGCs.

11) Subsection “Simultaneous removal of gcl and Sxl ameliorates the gcl phenotype” and Figure 5: The data would be more convincing if the authors had some independent way to classify male vs. female embryos, other than expressing or not expressing Sxl. Could they use a balancer with GFP or some other way to distinguish male control with intact Sxl from female embryos with Sxl deleted?

We have examined the rescued embryos using *sis-b* specific smFISH in addition to *Sxl* signal. *sis-b* served as an independent marker to assess if there is a sex-bias and whether female (XX) embryos are preferentially rescued, leading to a bipolar distribution. Since we examined blastoderm stage embryos, *Sxl* RNA in-situ allowed us to sex the embryos based on presence (female) or absence (male). We further confirmed the sexual identity by the presence of either two dots (2X i.e. female) or a single dot (1X i.e. male) of *sis-b* specific signal. We have included a representative example each of both a male and a female embryo illustrating this point in Figure 7 (newly added). This has allowed us to ascertain the genotype and the sex of the embryos unambiguously as recommended by the reviewer.

12) Subsection “A degradation-resistant form of Torso also activates transcription in PGCs”, and data in Figure 7: Again, please clarify the quantitation. Does 27% of torsodeg embryos express Sisb meant that all PGCs in that positive embryo have Sisb or at least one Pgc per embryo? The two have very different implications as to the strength of the effect. Might it be better to give counts on the percent of PGCs that score positive (counting at least 10 different embryos). What are the corresponding numbers for the wild type controls in this experiment? Panel A (wt) shows that some PGCs express Sisb. How significant is the difference between control and experimental for each?

Please see the response above for our rationale on quantifying embryos rather than individual buds/cells. We have clarified that only somatic nuclei in the WT embryos express *sis-b* as the nuclei in Figure 9A (renumbered from Figure 7) exhibiting transcription puncta are posterior somatic nuclei, not PBs.

13) Discussion: Could the effects on transcription be secondary to a failure to confine pole plasm components to the pole cell buds?

This is a good point and is in fact in agreement with our model that pertains to centrosome-mediated transport and sequestration is at the heart of establishment of germline/soma distinction in the early embryo. Future experiments will indeed focus on testing different aspects of the model (please refer to last two paragraphs of Discussion).

Reviewer #2:Past work suggests that the role of the Drosophila Germ Cell-Less (GCL) is to maintain primordial germ cells in a transcriptionally quiescent state. Pae et al., 2017 found that GCL mediates degradation of the Torso RTK, whose ectopic activity perturbs pole cell development. Colonetta et al., confirm that some somatic genes are ectopically activated in gcl mutant pole buds/cells and that ectopic GCL can inhibit expression of some genes in somatic cells, as can a mutant version of Torso (from Pae et al.,) that is not degraded by GCL. Contradicting Pae et al., Colonetta et al., argue that Torso's effects on PGCs require MAP Kinase pathway signaling. They also describe a novel phenotype in Torso[deg], gcl mutant, and MEK gain-of-function MEK embryos where pole plasm components get mis-localized. My criticisms are as follows: Several of the experiments contradicting/correcting Pae et al., are not definitive. Moreover, even if correct, the findings refine the model of Pae et al., without providing much new mechanistic insight. While of interest to Drosophila workers, particularly ones with an interest in early patterning and primordial germ cells, this manuscript might be considered esoteric by others who may also find reading difficult, owing to some poor writing.

In our estimate the question of ‘significance’ raised by reviewer #2 is the most crucial. Reviewer #2 suggests that elucidating the role of *gcl* in germline/soma distinction is of little interest outside a subset of the fly community. This seems to us to be an unusually narrow view. We believe that the central question addressed in our manuscript namely “to understand how *gcl* functions in establishing germline/soma distinction in early *Drosophila* embryos” is of broad relevance to stem cell biologists and those interested in gene regulation and early development. For example, many other invertebrates and even vertebrates utilize localized determinants for germline specification, and must encounter somewhat similar problems in germline/soma distinction.

In addition, *gcl* homologs are not only found in mammals, but also the mouse *mgCl^-^1* can rescue *gcl* mutants. Like the fly protein, mGCl^-^1 is associated with the nuclear matrix and it interacts directly with the inner nuclear membrane proteins LAP2β, Emerin, and MAN1. The LAP2β:mGCl^-^1 complex is reported to sequester E2F:D1 to the nuclear envelope, reducing E2F:D1 transcriptional activity. Although the *mgCl^-^1* gene is not required for germline specification, it does have a role in spermatogenesis. It is also required for normal nuclear morphology in liver and endocrine pancreatic cells. Given the functional conservation of mammalian Gcl proteins, their localization to the nuclear envelope, and their role in transcriptional regulation (newly added lines in Introduction), it seems likely that interest in the functions of the *Drosophila* protein would not be restricted to the fly community.

Also, in this context, the mass spec analysis in Pae et al., of proteins pulled down by WT Gcl is of interest. Near the top of the list of co-immunoprecipitated proteins are two components of the ‘facilitates chromatin transcription’ (FACT) complex. The FACT subunit Spt16 is second on the list with a score of 332. The other FACT subunit, Ssrp, is seventh on the list with a score of 231. Obviously, the presence of two transcription elongation factors is intriguing as sequestration of FACT in the nuclear envelope by Gcl could provide an additional (even if only a backup) mechanism for shutting off ongoing transcription.

Reviewer #2 also suggests that our paper is only a refinement of previously published work from the Lehmann lab. On this point, we also disagree.

One significant issue is transcription. An important contribution of Cinalli and Lehmann is, that contrary to previous studies by the Jongens lab, it “establishes” that transcription is not only irrelevant to *gcl* function during PGC formation but apparently is also incorrect. In the section of that paper that focuses on transcription, the authors state this point explicitly:

“Thus, we conclude that Gcl does not inhibit Pol II dependent transcription during PGC formation.”

In the Introduction, Pae et al., cite this earlier work as evidence for a non-transcriptional role of Gcl in PGC formation stating:

“…other experiments indicated that the major function of GCL is likely independent of transcriptional regulation (Cinalli, 2012; Cinalli and Lehmann, 2013).”

This same point is reiterated in the Results section, postulating that there is an alternative non-transcriptional pathway downstream of Torso that must be blocked by Gcl. They state:

“This is consistent with previous findings that the PGC formation defect seen in *gcl* embryos cannot be rescued by global transcriptional inhibition (Cinalli and Lehmann, 2013).”

Our findings clearly contradict the first claim that *gcl* does not inhibit PolII transcription. Our findings also contradict the second claim that silencing transcription is not an important *gcl* function.

To begin with, the observation that silencing *Sxl* is an important function for *gcl* in PGC formation is clearly not a “refinement” of the conclusions drawn in Cinalli and Lehmann and subsequently in Pae et al. They concluded exactly the opposite—that there were no relevant gene targets that needed to be silenced in PBs/PGCs by *gcl*.

Second, the studies of Pae et al., suggest that one (if not *the*) target for *gcl* in PGC formation/specification is the terminal pathway receptor Torso. They showed that Gcl interacts with Torso to promote Cul3-dependent degradation. In the absence of *gcl*, Torso is not degraded in newly formed pole buds (PBs), and, by an unspecified mechanism (that is postulated to be distinct from the canonical Ras-Raf-MEK-MAPK signaling cascade), the presence of activated Torso disrupts PB cellularization/specification. If this model is (generally) correct, then a degradation-resistant form of Torso should mimic key *gcl* phenotypes. One of the *gcl* phenotypes observed by us and by the Jongens’ lab is transcriptional. *gcl* PB nuclei fail to shut off ongoing transcription of X-chromosome counting elements and inappropriately turn on transcription of the *Sxl* establishment promoter, *Sxl-Pe*. We have shown that PBs/PGCs in *torso^deg^* embryos also express these genes. Thus, proteolysis-resistant Torso activates transcription in PBs/PGCs, including a key *gcl* transcriptional target. Again, this is not a “refinement” of the main conclusions of Pae et al., as this paper clearly stated that transcription isn’t relevant.

Finally, Pae et al., proposed that *gcl* mediated degradation of Torso blocked the activation of a non-canonical *and* non-transcriptional terminal signaling pathway. Since activated MEK didn’t turn-on transcription in PBs/PGCs, we can’t rule out the existence of a “non-canonical” Torso/Torso-like signal transduction cascade that is independent of MEK/ERK. However, in this case, this postulated MEK/ERK independent pathway must be responsible for inappropriately activating transcription in PBs/PGCs in *gcl* mutants. In particular, we find that removing *torso-like* not only rescues PGC formation in *gcl* embryos but also eliminates transcription in the *gcl* PBs/PGCs. This is also not a “refinement” of Pae et al.

A second relevant *gcl* phenotype, not discussed in Cinalli and Lehmann, or in Pae et al., is the misdistribution of the germ plasm. In WT, germ plasm (mRNAs and proteins) is initially anchored to the actin cortical cytoskeleton. When nuclei (or more specifically centrosomes) enter the posterior pole, this triggers the release of the germ plasm and its subsequent dynein/MT dependent transport and accumulation around PB nuclei. This process is disrupted in *gcl* embryos. As in WT, PB nuclei trigger the release of the germ plasm in *gcl* embryos; however, it is not properly distributed around the PB nuclei. While some germ plasm remains concentrated on the posterior side of the *gcl* PB nuclei, much of the germ plasm “escapes” into the surrounding soma. Our previously reported findings indicated that abnormalities in centrosome/MT behavior in *gcl* PBs are responsible not only for the loss of germ plasm but also for defects in PB cellularization. Significantly, we find that *Torso^deg^* elicits a similar spreading of the germ plasm into the soma, suggesting that the failure to degrade Torso also impacts centrosomes/MTs in PBs. This possibility was not considered by Pae et al.

Pae et al., suggest that there is a novel non-canonical pathway downstream of Torso that interferes with PGC formation. While this is a possibility, MEK/ERK have been reported to impact MT function. Consistent with the idea that the canonical kinase cascade needs to be shut down in PBs by *gcl*, we find that activated MEK induces a similar spreading of germ plasm as is observed in *gcl* mutants and in Torso^deg^. While this effect could be non-transcriptional, it nevertheless requires canonical terminal pathway components.

1) The authors clearly show a number of genes, particularly ones involved in sex determination (sis-b, runt, Sxl), are expressed ectopically in polar buds/primordial germ cells, in the absence of GCL. While the authors add some additional genes to the list, this phenomenon has previously been demonstrated by these and other authors, and as noted in this manuscript, although the use here of smFISH to detect nascent transcripts is a nice touch.

As the reviewer points out, the Jongens lab reported that several genes are inappropriately expressed in PGCs. However, the involvement of Gcl during establishment and/or maintenance of transcriptional quiescence was subsequently discounted by Cinalli and Lehmann, and Pae et al. This raised a critical question underlying the central function of Gcl: does it or does it not contribute to transcriptional quiescence required for germline/soma distinction? For this reason, it was important to determine which of the previously reported findings were correct. Our results confirmed findings from the Jongens lab. We extended these results by examining *Sxl-Pe* expression. Since we found that disrupting *Sxl* function partially rescued the PGC formation/specification defects in *gcl* embryos, an obvious prediction is that *Sxl-Pe* is inappropriately activated in *gcl* PB/PGCs. Thus, this experiment needed to be done as well.

2) The authors express gcl ectopically at the anterior of the embryo, showing that this leads to a local reduction in the expression of Sxl and use this finding to argue that GCL inhibits transcription of specific targets, and is sufficient to do so, "even in the absence of other pole plasm components," presumably including Torso. However, Torso is active in this domain and if Torso does lead to the activate of transcription Sxl via its effects upon sis-a, sis-b, runt, then this might be the expected result. If indeed the effect of Torso on pole cells is transcriptional, then Torso may be the sole target of GCL. The definitive experiment to answer this question has not been carried out by these authors and that is to assess the transcription of sis-a, sis-b, runt, Sxl, in pole buds/cells of embryos that are both null mutants for gcl and torso. If all of those genes failed to be expressed in the double mutant background, this would provide support for the notion that Torso acts to mediate the transcription of those genes, and might be the sole target of GCL. If any of those genes were expressed, that would argue that GCL can act to silence at least some genes, independent of Torso. Given the close chromosomal linkage of gcl and torso, it would be difficult, though not impossible, to obtain double mutants to test. It is certainly possible that with a concerted effort, these double mutants could be isolated and examined within a couple of months. Although the gcl tor/gcl tor double mutant would be ideal, the gcl/gcl; tsl/tsl strain might suffice, although we really do not have a complete picture of Tsl's role in the activation of Torso.

This is an excellent and very valuable suggestion. We thank the reviewer for emphasizing the importance of this experiment.

As the reviewer correctly pointed out, generating a recombinant between *gcl* and *torso* mutations will be arduous because of their proximity. However, Pae et al., used a *gcl/gcl; tsl/tsl* fly strain to demonstrate the rescue. We have successfully reproduced the results reported by Pae et al., i.e. embryos derived from *gcl/gcl; tsl/tsl* females showed a substantial rescue of the *gcl* PGC formation defects.

Not discussed in Pae et al., was whether the rescue of the PGC formation defects by eliminating *torso-like* in *gcl* embryos also results in the reestablishment of transcriptional quiescence. It does. Taken together with the finding that there is at least one gene, *Sxl*, that needs to be shutoff in PBs/PGCs, this observation provides strong support for the idea that one of the important functions of *gcl* in PGC formation is silencing transcription. Moreover, this function depends upon *gcl* promoting the degradation of the Torso receptor.

3) With regard to a transcriptional effect of Torso on pole buds/cells, the authors subtly misrepresent Pae et al. to set up a straw man in the statement that Pae et al., "proposed that activated Torso must inhibit PGC via distinct non-canonical mechanism that is both independent of the standard signal transduction and does not involve transcriptional activation." Although Pae et al., do state at several points in their paper that this effect of Torso is not transcriptional in nature, at the end of the manuscript they acknowledge that "additional downstream pathways (to MAPK/ERK) may require ligand-induced Torso receptor activation. Candidates include the JAK/STAT pathway, which is activated in dominant gain-of-function alleles of Torso (Li et al., 2002)." This would clearly be a transcriptional effect. Moreover, there is good reason to suspect that STAT may play a role in Torso's effects on pole buds/cells. First, there are several other contexts in which RTK's activate STAT. More to the point, however, the JAK/STAT pathway has been implicated in the control of Sxl transcription (Sefton et al., 2000), the mRNAs encoding the JAK/STAT pathway components are maternally loaded into the embryo, and Torso-dependent activation of STAT and Ras at an early step in PGC development has been shown to be required for their proliferation and migration (Li et al., 2003). Some of these reports should be of interest to Colonetta et al.

Our read of Cinalli and Lehmann as well as Pae et al. is somewhat different from that of the reviewers. According to these authors *gcl*’s role in transcriptional quiescence was, at best, inconsequential. Moreover, in some instances (see first quote above), they go farther and state that *gcl* has no role in downregulating transcription. Thus, when Pae et al., suggested that the JAK/STAT pathway might be one of the non-canonical pathways activated by Torso, they clearly did not mean that a JAK/STAT-dependent transcriptional output is relevant for PB/PGC formation/specification. Instead, they were likely referring to the reported defects in PGC mitosis, adhesion and migration when STAT activity is manipulated (Li and Li, 2003; Li et al., 2003).

As for the role of JAK/STAT in activating *Sxl-Pe*, the reviewer is correct: *Sxl-Pe* is a target of the JAK/STAT pathway. This was shown not only by Sefton et al., but also by Jinks et al., (2000) and Avila and Erickson, (2007). Jinks et al. showed that the effects of compromising the JAK/STAT pathway on *Sxl-Pe* transcription were only seen in the center of the embryos, while *Sxl-Pe* activity at the termini appeared to be normal.

The reason for the spatially restricted effects were investigated by Avila and Erickson, (2007). They also found that loss of zygotic *upd* and maternal *hop* (JAK) impacted *Sxl-Pe* in the center of the embryo, not at the termini. A somewhat different result was obtained in embryos lacking maternal *mrl* (STAT). Unlike *upd* or *hop*, a reduction in *Sxl-Pe* could be detected at the termini of the embryo when there was no maternal *mrl* (STAT). Based on the studies of Li and Li, and Lie et al., Avila and Erickson concluded that STAT activation by the Torso receptor likely accounted for the relative resistance of terminal somatic nuclei to a disruption in the *upd*-JAK dependent activation of *Sxl-Pe*.

However, there are other important findings in Avila and Erickson. First, the JAK/STAT pathway is not a key regulator of *Sxl-Pe*. To begin with, *upd* expression is first detected in nuclear cycle 13. By contrast, *Sxl-Pe* comes on in nuclear cycle 12 (according to Avila and Erickson, while we detected transcripts at nuclear cycle 11 using smFISH). Consistent with its expression pattern, *upd* mutations only impact *Sxl-Pe* activity in nuclear cycle 14, while earlier expression of *Sxl-Pe* resembles wild type. The same temporally restricted effects were observed for embryos lacking maternal *hop* (JAK) or *mrl* (STAT). Expression resembled wild type in nuclear cycles 11-13, while mutant embryos show defects in *Sxl-Pe* transcription only during nuclear cycle 14. Based on these findings, Avila and Erickson concluded that the JAK/STAT pathway is not needed for the initial activation of *Sxl-Pe* in the soma during nuclear cycles 11-13, but rather is required to sustain high levels of *Sxl-Pe* expression in nuclear cycle 14. Potentially supporting a role in maintaining transcriptional activity, but not the initial sex-specific activation of *Sxl-Pe*, Jinks et al., found that constitutively active JAK Kinase (Hop-Tum) does not induce *Sxl-Pe* in male somatic nuclei.

If the conclusions of Avila and Erickson are correct, activation of STAT in *gcl* PGCs is unlikely to be responsible for *Sxl-Pe* transcription in PB/PGC, let alone for the continued expression of other genes such as *sis-b*, *sis-a*, or *runt*. In addition, while Avila and Erickson found that *upd*/*hop* (JAK)/*mrl* (STAT) are only needed for *Sxl-Pe* activity in nuclear cycle 14, this is much later than the PB/PGC formation defects first manifest themselves in *gcl* embryos. It is also after the initial activation of *Sxl-Pe* in PB of *gcl* mutant embryos.

4) The section of the paper on "Simultaneous removal of gcl and Sxl ameliorates the gcl phenotype" is poorly written, probably impenetrable to anyone outside of the community of Drosophila workers studying early patterning events. The discussion of the expected progeny and the interpretation of the results are altogether unclear. Similarly, the experiment in which RNAi directed against Sxl is performed in the absence of GCL reveals an unexpected bimodal distribution of almost normal and pole cell-lacking embryos, for which no potential explanation is offered. In the absence of a credible potential explanation of this result, the conclusion that the loss of SXL is suppressing the lack of GCL is not convincing.

This section has been revised for clarity. Additionally, we have offered a plausible explanation for the bimodal distribution in the RNAi knockdown experiment. As we observed a modest female-specific bias with respect to ectopic activation of *Sxl* in early PGCs, we wondered if mitigating Sxl levels preferentially rescued PGCs in *gcl* female embryos as compared to males. As can be seen in Table 2 this turned out to be indeed correct. Altogether these data document a sex-specific nature of the rescue in the RNAi knockdown experiment and provide a reasonable explanation for its bimodal nature.

5) To ask the question of whether Map Kinase signaling downstream of Torso activity is required for Torso's effect upon pole buds/cells, Pae et al., performed RNAi against dsor1 (MEK) and rolled (MAPK) in a gcl null background. The failure of the RNAi treatment to suppress the effect of GCL loss on pole cells, was used as evidence that MAPK signaling downstream of Torso was not required for the Torso-dependent effects on pole buds/cells. To address this question, Colonetta el al., expressed activated MEK in mothers and assessed Sxl expression. Consistent with Pae et al., activated MEK failed to drive Sxl expression. However, they did observe Sxl expression in the somatic cells of some males. Given that this is a contrived and very different situation than pole cells, and that Pae's experiments in pole cells tested for requirement, while Colonetta's tested for sufficiency (apples to oranges) the question of MAPK signaling in Torso's effect upon pole buds/cells remains unanswered and it is baffling why Colonetta et al., did not examine sis-a, sis-b, runt, Sxl expression in the pole buds/cells of gcl mutant, dsor1 (MEK) and rolled (MAPK) knockdown embryos.

As noted above, the reviewer is correct in stating that activated MEK is unable to activate transcription. This is consistent with the idea that canonical downstream pathway components may not be involved in *transcriptionalactivation*. On this account, our data would potentially be in agreement with Pae et al. This was explicitly acknowledged in the text. We also pointed out that there were reasons to think that the levels of canonical terminal pathway signaling in PBs/PGCs might not be equivalent to that in *torso^Deg^* or *gcl* mutants when GOF MEK proteins are ectopically expressed. If that were the case, Torso/Torso-like dependent transcriptional activation in *gcl* PBs/PGCs might actually be mediated by the canonical signaling pathway. This issue will require additional studies, beyond the scope of the present manuscript.

6) Figure 9, Figure 10, Figure 11, Figure 12, and Figure 13 show plot profiles of various pole plasm components (Vasa, gcl RNA, pgc RNA, nos RNA) in individual wild-type, gcl mutant and Torso-deg embryos. While the photographic images of the pole plasm components are consistent with the interpretations proposed by the authors, the plot profiles, which apparently only show a subset of the examined embryos for each component/genotype are more effective in displaying embryo to embryo variation, that they are in showing consistent patterns of mis-localization associated with the pole plasm components in the three genetic backgrounds.

We agree with the reviewer that these phenotypes are variable. The plots show embryo-to-embryo variability within a single genotype, but more importantly, they also serve to demonstrate between genotypes. For example, Figure 11 shows the distribution of Vasa is similarly skewed in *gcl* and *Torso^deg^* mutants resulting in higher levels of Vasa far from the posterior pole. This response is absent in WT controls. However, the distinction between the WT and the experimental embryos is clear-cut. Also, these phenotypes for the first time establish that Torso receptor and the downstream components of the pathway can affect RNA/protein transport that is likely mediated by MT centrosomes. We believe that this is an important finding with implications that go far beyond *Drosophila* germ cell biology.

[Editors’ note: what follows is the authors’ response to the second round of review.]

It is our judgment now that the manuscript likely warrants publication in eLife if you can address the following remaining concerns raised by one of the reviewers:1) The distinction between terminal and non-terminal nuclei with respect to the ability of ectopic anterior Gcl to inhibit Sxl transcription should be explained more fully. All of the instances of ectopic Sxl transcription reported here occur in situations in which Torso or MEK signaling are occurring. That being the case it is not possible to conclude unambiguously that Gcl is uniquely able to suppress Sxl transcription. It's ability to suppress Sxl transcription may rely on Sxl transcription having been activated by ectopic Torso or MEK activity. While I would not require the authors to carry out additional experiments in which Gcl is expressed outside of the terminal regions of wild-type female embryos, or even better, in Torso-lacking female embryos, I believe that it is at least essential to discuss this issue and how it affects the conclusion that Gcl is uniquely able to suppress Sxl transcription. On the other hand, the results of examining Sxl transcription in Torso mutant-derived female embryos ectopically expressing Gcl might prove illuminating, if the authors are willing to undertake the effort.

We have altered the text to account for both possibilities (please refer to the Discussion). We are not able to unambiguously attribute transcription inhibition to direct action by Gcl with our experiments. However, it is worth noting that in our experiment ectopically localizing *gcl* to the anterior (*gCl^-^bcd-3’UTR*), *Sxl* expression should be predominantly under the control of the canonical somatic pathway, namely activation by X-linked counting elements (Salz and Erickson, 2010), rather than a Torso-mediated pathway. While Torso is present at the anterior of the embryo, *Sxl* is activated throughout the soma, even in the absence of Torso in the middle of the embryo. Therefore, ectopic Gcl may be able to repress *Sxl* transcription activated by a Torso-independent pathway, though we are not able to confirm this since Torso is present in the anterior terminus of the embryo as well as the posterior terminus that we focus on for much of the manuscript. On the other hand, our results indicating that *tsl* mutations can rescue the PGC transcription phenotype of *gcl* mutants suggest that the PGC transcription we observe likely results from an inability of Gcl to directly inhibit the Torso pathway in the PGCs.

Unfortunately, *torso* null mutants are female sterile, making the proposed experiment extremely challenging (Schupbach and Wieschaus, 1989).

2) The means of expression of the various transgenic construct used should be explained more clearly with respect to whether the constructs are expressed in the female under Gal4-mediated control and deposited in the embryo versus maternal deposition of Gal4 in the embryo and zygotic expression of the gene that has been crossed in. Similarly, the identities of the transgenic constructs, UASt, UASp, or UASother-based, should be noted.

We have altered the text to add details of how transgenes were expressed throughout. Please see experimental descriptions in the Results for zygotic expression, combination of maternal and zygotic depletion for *gcl* and *Sxl*, respectively, zygotic expression, maternal depletion of both *gcl* and *tsl*, maternal expression for clarification on individual experiments. Details of how these genes were expressed are also in the legends of the figures.

Additionally, we have added details of the UAS constructs in the reagents table of Materials and methods. All UAS constructs are expressed in the germline since they are either UASp or VALIUM20 (which the BDSC website specifies works in both soma and germline).

3) The section in the Discussion on the effects of activated MEK, and of the ERK/MAPK cassette on Sxl transcription in PBs/PGCs versus somatic cells should be modified with additional consideration of the possibility that an alternative pathway downstream of Torso does exist, possibly working independent of the MAPK/ERK pathway. In modifying the text, see the more detailed consideration of this possibility in the Public Review, which outlines my reasoning.

We have modified the text, as suggested. Please see the Discussion for our statement considering alternative downstream pathways.